# Learning Auctions with Robust Incentive Guarantees

**Jacob Abernethy**
Georgia Tech
prof@gatech.edu

**Rachel Cummings**
Georgia Tech
rachelc@gatech.edu

**Bhuvesh Kumar**
Georgia Tech
bhuvesh@gatech.edu

**Jamie Morgenstern**
Georgia Tech
jamiemmt.cs@gatech.edu

**Samuel Taggart**
Oberlin College
sam.taggart@oberlin.edu

## Abstract

We study the problem of learning Bayesian-optimal revenue-maximizing auctions. The classical approach to maximizing revenue requires a known prior distribution on the demand of the bidders, although recent work has shown how to replace the knowledge of a prior distribution with a polynomial sample. However, in an online setting, when buyers can participate in multiple rounds, standard learning techniques are susceptible to *strategic overfitting*: bidders can improve their long-term wellbeing by manipulating the trajectory of the learning algorithm through bidding. For example, they may be able to strategically adjust their behavior in earlier rounds to achieve lower, more favorable future prices. Such non-truthful behavior can hinder learning and harm revenue. In this paper, we combine tools from differential privacy, mechanism design, and sample complexity to give a repeated auction that (1) learns bidder demand from past data, (2) is approximately revenue-optimal, and (3) strategically robust, as it incentivizes bidders to behave truthfully.

## 1 Introduction

When we observe prices in market settings—stock exchanges, farmers' markets, ad auctions—we understand that these prices were not chosen arbitrarily. Rather, the seller (auctioneer, market maker, etc.) selected these prices after observing a stream of previous transactions, which provide relevant information about the demands of buyers that are key to maximizing income as well as managing available inventory. The process of setting prices from a growing database of previous sales is fundamentally a learning problem, with all of the typical tradeoffs akin to bias versus variance, etc. In the case of repeated auctions, however, there is one additional challenge: market participants are often quite aware of the underlying learning procedures employed by the auctioneer and can seek to benefit using deceptive bidding strategies. Buyers, in other words, can aim to *induce* overfitting, introducing additional hurdles to learning problem at hand.

Under bayesian assumptions, and in a batch setting where agents only act once, auction pricing has been well-understood since the work of Myerson [39], who characterized the revenue-optimal scheme as a function of the prior distribution of values of the bidders. Frequentist alternatives to this model have been introduced in recent years [21, 13, 4, 18, 37, 5, 38, 25, 17, 23], with the goal of designing auctions with good revenue guarantees if one does not have a prior but instead is given only samples from the underlying distribution. These methods, however, still imagine only a one-shot mechanism and are not robust to multi-round strategic behavior of bidders.

This paper studies the design of multi-round auction-learning algorithms that exhibit theoretical guarantees that limit a buyer's ability to manipulate the mechanism towards their own benefit. Our

results aim to nudge the development of optimal auctions closer to realistic environments where such mechanisms are deployed. We employ tools from *differential privacy* as our core technique to control the impact of any individual buyer's strategy on her utility in future participation. A differentially private mechanism ensures that that the output of a computation has only a small dependence on any one input data point. Privacy has been previously used as a tool to achieve truthfulness in a variety of game theoretic environments [14], including mechanism design [33, 40], mediated games [29, 16], and market design [15, 28, 44, 12]. Our seller's learning algorithm is differentially private with respect to bid data, which limits the effect of each player's bid on future choices of single-round auctions, thus disentangling incentives across rounds. In this sense we use differential privacy not as a tool for information security but instead for robustness; this in turn yields the desired incentive guarantees.

Our two main results are the first computationally tractable algorithms for learning nearly-optimal Bayesian revenue-maximizing auctions with an approximate truthfulness guarantee. We first give a learning algorithm for which there exists an approximate equilibrium in which all bidders report their true values. Along the way to this result, we provide several useful technical lemmas for comparing the revenue of mechanisms on two similar distributions, and comparing the revenue of two similar mechanisms on any fixed distribution. These may find future applications of independent interest beyond this work. Second, under an assumption limiting changes in buyer behavior from round to round, we construct a mechanism for which there is an exact equilibrium where every bidder bids within a small constant of their value, which also achieves a nearly optimal revenue guarantee. A mechanism with this guarantee is substantially more complex to achieve, requiring arguments about what bidders might learn about each other across rounds.

**Related Work**   The classical Myerson auction [39] maximizes revenue when selling a single item to buyers drawn from a fixed, known distribution. More recent work investigated how to maximize revenue when that distribution is unknown but some small number of samples are available [21, 13, 4, 18, 37, 5, 38, 25, 17, 23] (or in an online model [9]). This line of work assumes every buyer in the sample reported their values *truthfully*, rather than manipulating their reported data. If each buyer appears in at most one auction, then they will not have incentive to misreport their bids. However, strategic buyers participating repeatedly in these auctions may have incentive to manipulate their behavior so as to guarantee themselves more utility in future rounds. We extend this line of work by assuming each buyer can participate in several auctions, and so we must analyze bidders' incentives across rounds if we hope to learn a good auction from their past bidding behavior. Related line of work includes dynamic auctions [42, 3, 36, 31] where the buyers are strategic about how their behaviour today effects the seller's behaviour in the future and [8] where the buyer uses a non regret learning algorithm to bid across the rounds.

Our results are related to work on iteratively learning prices [7, 35, 10, 41], although these results do not consider multiple appearances of bidders across rounds. Most closely related to our work is that of Liu et al. [30], which assumes buyers may appear more than once, and finds no-regret posted prices or anonymous reserves. We leverage several of their novel ideas, such as maintaining a differentially private internal state to guarantee approximate truthfulness. Further, our work optimizes over a substantially more complicated space of mechanisms.

With repeated appearances of each buyer, our auction learning problem comes to resemble dynamic mechanism design. We therefore review some of the relevant literature. A truthful mechanism is given in [6] that exactly maximizes social welfare in a dynamic environment, and [27, 43] extended this mechanism to maximize revenue. In contrast, our mechanism approximately maximizes revenue in a dynamic environment with much looser assumptions on buyers' value distributions, but compares to the weaker per-round benchmark of the optimal single-shot revenue. Epasto et al. [22] takes a similar approach and proposes a dynamic mechanism for buyers who strategize across rounds under a *large market* assumption. In contrast to our proposed mechanism, theirs will not generally run in polynomial time. Additionally, we focus on mechanisms which *explicitly* limit an agent's ability to manipulate them.

## 2   Model and Preliminaries

**The model**   We consider a $T$-round auction, where a seller sells a supply of $J$ identical items to $n$ unit-demand bidders each round. For each round $t$ and population $i$ a value $v_{i,t}$ is sampled from

a fixed distribution $D_i$, representing the amount the bidder is willing to pay for the item. We let $\mathbf{D} = D_1 \times \cdots \times D_n$ denote the product distribution of value distributions, and we use $\mathbf{v}$ to denote a vector of values sampled from this distribution. Further, we let $\mathbf{v}_{-i}$ denote $\mathbf{v}$ with the $i$-th element removed, and use $(v_i', \mathbf{v}_{-i})$ to denote the same vector with $v_i'$ replacing the $i$-th element.

We consider a setting similar to Liu et al. [30] where a bidder from any population may appear several times over the course of the $T$ rounds, drawing a fresh value each time.[1] In this setting, bidders may have an incentive to misreport their values in order to change the mechanism in future rounds, and their potential reward for doing so depends on the number of future rounds in which they expect to participate. Amin et al. [1] show that very little can be done when a bidder participates in every round, so we assume this cannot occur. Formally:

**Assumption 1.** *No bidder participates in more than $k$ rounds of the $T$-round auction.*

**Mechanism Design Basics**  One can view a *mechanism* (auction) $\mathcal{M} := (x, p)$ as having two components: (a) a possibly-randomized *allocation rule* $x : V^n \to \mathcal{X}$, which takes in a vector of values (bids) $\mathbf{v}$ and returns a feasible allocation of the items, where $x_i(\mathbf{v})$ is 1 if $i$ receives the item and 0 otherwise; and (b) a *payment rule* $p : V^n \to \mathbb{R}^n$, which takes $\mathbf{v}$ and outputs a vector of payments demanded of each player. Assuming, for the moment, that bidders bid their true values $\mathbf{v}$, we can define the *expected revenue* of a mechnamism $\mathcal{M}$ as the expectation of the payments received,

$$\mathrm{Rev}(\mathcal{M}; D) := \mathbf{E}_{\mathbf{v} \sim \mathbf{D}}[\sum_{i=1}^n p_i(\mathbf{v})].$$

We make the standard assumption that the participants have *quasi-linear utility*: for a vector of bids $\mathbf{v}'$ (which may not necessarily match the values $\mathbf{v}$), a bidder's utility for allocation $x_i(\mathbf{v}')$ and payment $p_i(\mathbf{v})$ is

$$u_i(x, p, \mathbf{v}') = v_i \cdot x_i(\mathbf{v}') - p_i(\mathbf{v}').$$

We may now introduce the notion of a *truthful* mechanism. A mechanism $\mathcal{M} := (x, p)$ is deemed truthful if, given a vector of true values $\mathbf{v}$ and some arbitrary vector of bids $\mathbf{v}'$, each agent receives no less utility bidding $v_i$ rather than $v_i'$; that is, for every $i$, it must hold that $u_i(x, p, \mathbf{v}') \le u_i(x, p, (v_i, \mathbf{v}_{-i}'))$.

Let us now recall a classical result in Bayesian-optimal mechanism design when the seller's goal is to maximize revenue. Myerson [39] essentially fully characterized the solution in this setting. The interested reader can learn more in Hartline [24]; we briefly review these results here in two pieces. The first piece states that payments in truthful mechanisms essentially depend solely on the allocation function.

**Theorem 2** (Payment Identity, Myerson [39])**.** *A mechanism is truthful if and only if it has a monotone allocation rule and payments which for all valuation profiles $\mathbf{v}$ satisfy*

$$p_i(\mathbf{v}) = v_i \cdot x_i(v_i, \mathbf{v}_{-i}) - \int_0^{v_i} x_i(u, \mathbf{v}_{-i}) du + p_i(0, \mathbf{v}_{-i}).$$

The second key result is that for truthful mechanisms, the expected revenue can be written in terms of welfare in a remapped *virtual value* space.

**Theorem 3** (Myerson [39])**.** *For any truthful mechanism $\mathcal{M} = (x, p)$ with values distributed according to $D$, the expected revenue from player $i$ can be written as $\mathbb{E}_{\mathbf{v} \sim D}[\phi_i(v_i) x_i(\mathbf{v})]$, where $\phi_i(v_i)$ is the* virtual value, *given by $\phi_i(v_i) = 1 - \frac{1 - F_i(v_i)}{f_i(v_i)}$. So, $Rev(\mathcal{M}; D) = \mathbb{E}_{\mathbf{v} \sim D}[\sum_i \phi_i(v_i) x_i(\mathbf{v})]$.*

We will use the notation $\mathcal{M}_{\mathbf{D}}^*$ to denote the revenue-optimal mechanism for distribution $\mathbf{D}$— Myerson provides a precise construction of this auction.

**Definition 4** (Myerson's Auction)**.** *Fixing a prior distribution $\mathbf{D}$, given a value profile $\mathbf{v}$ Myerson's revenue-optimal mechanism $\mathcal{M}_{\mathbf{D}}^*$ calculates virtual values $\phi_i(v_i) = v_i - \frac{1 - F_{D_i}(v_i)}{f_{D_i}(v_i)}$ and (a) selects the feasible allocation which maximizes virtual welfare according to the virtual values and (b) charges payments according to the Payment Identity of Theorem 2.*

**Truthfulness and Dynamic Equilibrium**   The mechanism design preliminaries discussed previously are for *one shot* games where players do not observe past actions of others and adjust their strategy accordingly. We now turn our attention to multi-round play, we need to expand our notion of player behavior and strategy. We will now assume that a strategy for each bidder maps the history of observed actions to contingency plans over actions in the current and future rounds: we assume agents observe their own outcomes $x_i^t$ and $p_i^t$ in rounds in which they participate, but not the full historical data used by the designer to produce the mechanism each round. A history $h_i^t$ in round $t$ for each agent $i$ then consists of a bid, an allocation, and a payment for each round the agent has participated in. Given a history $h_i^t$, and value $v_i^t$ in the current round $t$ for agent $i$, we denote agent $i$'s strategy as $\sigma_i^t(v_i^t ; h_i^t)$. We suppress the dependence on the history when clear from context.

Note that because agents do not observe the historical bids of others, they must form probabilistic beliefs about these bids, and we assume these beliefs are Bayesian in nature. Denote by $\mu_i^t(h_i^t)$ the beliefs of agent $i$ in round $t$ after observing history $h_i^t$. A profile of strategies is an *equilibrium* for this game if in every round $t$, every agent $i$, and every history $h_i^t$ that agent $i$ might have observed previously, agent $i$'s strategy maximizes their expected total utility over the current and future rounds; the expectation is taken over the randomness of agent $i$'s beliefs as well as the future values of all agents. Formally, let $U_i^t(\sigma, v_i^t, h_i^t)$ denote the total expected utility of agent $i$ in rounds $t, t+1, \ldots, T$ given a value of $v_i^t$ in the current round and an observed history $h_i^t$, playing strategy $\sigma$.

**Definition 5** (Perfect Bayesian Equilibrium (PBE)). *A profile of strategies $\boldsymbol{\sigma}$ is an $\eta$-approximate Perfect Bayesian equilibrium if for every agent $i$, round $t$, history $h_i^t$ for agent $i$ in round $t$, value $v_i^t$ for $i$ in round $t$, the strategy $\sigma_i^t(v_i^t ; h_i^t)$ approximately maximizes agent $i$'s total expected utility from future rounds up to an additive $\eta$. That is, $U_i^t(\sigma_i^t(v_i^t ; h_i^t), v_i^t, h_i^t) \geq U_i^t(\sigma', v_i^t, h_i^t) - \eta$ for every alternate strategy $\sigma'$. If $\eta = 0$, we say that $\boldsymbol{\sigma}$ is an* exact Perfect Bayesian Equilibrium.

In other words, a PBE is a strategy $\boldsymbol{\sigma}$ such that, for every bidder $i$ and for every history of the game, if all other bidders besides $i$ behave according to the strategy $\boldsymbol{\sigma}$, then playing $\sigma_i$ is approximately utility-maximizing behavior for bidder $i$.

**Definition 6** ($\eta$-utility-approximate BIC). *A mechanism is $\eta$ utility-approximately Bayesian incentive compatible if the strategy profile where every agent bids truthfully in every history is an $\eta$-approximate Perfect Bayesian equilibrium.*

We also consider a more robust notion of incentive compatibility, where there exists an exact equilibrium with each agent bidding $\eta$-close to her true value, a notion also used in Liu et al. [30].

**Definition 7** ($\eta$-bid-approximate BIC). *A mechanism is $\eta$ bid-approximate BIC if $\exists$ an exact PBE where each bidder bids within $\eta$ of their value in every history.*

Definition 6 guarantees that all bidders bidding truthfully in all rounds is an (approximate) Bayes-Nash equilibrium (BNE). In proving a mechanism utility-approximate BIC, one therefore may assume bidders report truthfully in future rounds. Consequently, the only impact an agent's bid has on their future utility is through their impact on future mechanisms. Definition 7 guarantees the existence of an exact equilibrium in which all bidders bid within $\eta$ of their value. Therefore, mechanisms with this guarantee will need to ensure bidders do not change their behavior in later rounds to influence other bidders' behavior in earlier rounds.

**Differential Privacy Background**   We now provide some basics on differential privacy, our main technique that helps guarantee approximate truthfulness in equilibrium. We refer to a *database* $Z \in \mathcal{Z}^n$ as a collection of data from $n$ individuals, and we say that two databases are *neighboring* if they differ in at most one entry.

**Definition 8** (Differential Privacy [19]). *An algorithm (mapping) $\mathcal{A} : \mathcal{Z}^n \to \mathcal{R}$ is $(\epsilon, \delta)$-differentially private if for neighboring databases $Z, Z' \in \mathcal{Z}^n$ and subsets of possible outputs $\mathcal{S} \subseteq \mathcal{R}$, we have $\mathrm{P}[\mathcal{A}(Z) \in \mathcal{S}] \leq \exp(\epsilon) \, \mathrm{P}[\mathcal{A}(Z') \in \mathcal{S}] + \delta$.*

The parameter $\epsilon$ quantifies the algorithm's privacy guarantee; smaller $\epsilon$ corresponds to stronger privacy. A key property of differential privacy is that it is robust to *post-processing*.

**Lemma 9** (Post-processing [19]). *Let $\mathcal{A} : \mathcal{Z}^n \to R$ be an $(\epsilon, \delta)$-differentially private algorithm and let $f : \mathcal{R} \to \mathcal{R}'$ be a random function. Then $f \circ \mathcal{A} : \mathcal{Z}^n \to \mathcal{R}'$ is also $(\epsilon, \delta)$-differentially private.*

We need a more precise notion of privacy when multiple agents are involved receiving different information. For notation, we say two databases are *i-neighbors* if they differ only in the $i$-th entry. Also, let $\mathcal{A}(Z)_{-i}$ denote the vector of outputs to all players except player $i$.

**Definition 10** (Joint Differential Privacy [29]). *An algorithm* $\mathcal{A} : \mathcal{Z}^n \to \mathcal{R}^n$ *is* $(\epsilon, \delta)$-*jointly differentially private if for every* $i \in [n]$, *every pair of i-neighbors* $Z, Z' \in \mathcal{Z}^n$, *and every* $\mathcal{S} \subseteq \mathcal{R}^{n-1}$,

$$\mathrm{P}[\mathcal{A}(Z)_{-i} \in \mathcal{S}] \leq \exp(\epsilon) \mathrm{P}[\mathcal{A}(Z')_{-i} \in \mathcal{S}] + \delta.$$

Finally, we can reason about the joint differential privacy of mechanism decomposed into a public sanitized broadcast, i.e. as if on a "billboard," and a private non-sanitzed portion. The following lemma shows that privacy is still preserved under such a decomposition.

**Lemma 11** (Billboard Lemma [26]). *Suppose* $\mathcal{A} : \mathcal{Z}^n \to \mathcal{R}$ *is* $(\epsilon, \delta)$-*differentially private. Consider any collection of functions* $f_i : \mathcal{Z}_i \times \mathcal{R} \to \mathcal{R}'$, *for* $i \in [n]$, *where* $\mathcal{Z}_i$ *is the portion of the database containing i's data. Then the composition* $\{f_i(\Pi_i(Z), \mathcal{A}(Z))\}$ *is* $(\epsilon, \delta)$-*jointly differentially private, where* $\Pi_i : \mathcal{Z} \to \mathcal{Z}_i$ *is the projection to i's data.*

The final tools we borrow from differential privacy are the exponential mechanism [34], and the ability to maintain a histogram estimate of values which arrive in one at a time. The primary technique for the latter involves data structures known as tree-based aggregations [20, 11]. This protocol is a differentially private method for calculating the cumulative sum of elements from 1 to $t$ for any $t \leq T$, for which at any round $t$ the protocol can return an estimate of the number of elements prior to round $t$, for which the entire execution is differentially private. We provide more details about our instances of these algorithms in Appendices B.1.3 and A.1.1.

# 3 Utility-Approximate Bayesian Incentive Compatibility

In this section, we give an online algorithm (Algorithm 1) for learning the optimal auction which is utility-approximate BIC. The main idea is to use differential privacy to explicitly control the amount of information the auctioneer takes forward from round $t$ to later rounds. We do so by maintaining a differentially private estimate $H'_{i,t}$ of each empirical bid distribution, and choosing future auctions based only on this differentially private estimate. Thus, from the perspective of any bidder, her behavior in round $t$ has very little chance of affecting any of the auctions selected in later rounds. In round $t$, we run Myerson's mechanism with prior $H'_{i,t}$ to compute allocations and payments. Thus the one-shot mechanism in round $t$ is exactly incentive compatible with respect to the current round. The differentially private subroutine used to compute estimate $H'_{i,t}$ is described in full detail in Appendix A, with privacy, truthfulness, and revenue guarantees presented in Section 3.1.

---

**Algorithm 1:** Utility-Approximate BIC Online Auction

---
**Parameters:** discretization $\beta$, privacy $\epsilon$, upper bound on support $h$, num. of rounds $T$
**Initialize:** $H'_{i,0} \leftarrow \text{Uniform}(0, h)$ for $i = 1, \cdots, n$
**for** $t = 1, \cdots, T$ **do**
    Receive bid profile $\mathbf{v}_t = (v_{1,t}, \ldots, v_{n,t})$, rounded down to integer multiple of $\beta$
    Run Myerson (Def. 4) with $\mathbf{H'}_{t-1}$ as prior and $\mathbf{v}_t$ as bid for allocations/payments.
    **for** $i = 1, \ldots, n$ **do**
        Update $H'_{i,t}$ via two-fold tree aggregation (Algorithm 3), giving as input $v_{i,t}$
    **end**
**end**

---

## 3.1 Privacy, Truthfulness, and Revenue Guarantees

We now prove that the learning subroutine of Algorithm 1 satisfies differential privacy (Theorem 12), use this to prove the mechanism is utility-approximate BIC (Theorem 13), and show that Algorithm 1 achieves a $o(1)$ additive approximation to the optimal revenue (Theorem 14).

**Theorem 12.** *The stream of estimates* $\{\mathbf{H'}_t\}_{t=1}^T$ *maintained by Algorithm 1 is* $(\epsilon, \epsilon/T)$-*differentially private with respect to the stream of input bids* $\{v_t\}_{t=1}^T$.

We emphasize that Theorem 12 does not claim that Algorithm 1 is itself differentially private, it only states that the procedure rests on a differentially private subroutine. This distinction is critical: our

algorithm is not differentially private in its selection of allocations and payments in round $t$. However, the information the mechanism carries forward (namely, the estimated empirical distribution) is maintained in a differentially private manner. This is sufficient for guaranteeing that bidders' behavior in round $t$ does not significantly affect which auctions are selected in later rounds. This will allow us to prove a utility-approximate BIC guarantee, but will not be sufficiently strong to argue about bid-approximate BIC. For that, we will need to additionally ensure that the allocations (and payments) in round $t$ are differentially private (jointly differentially private), see Section 4.

We now turn our attention to proving a guarantee on the truthfulness of Algorithm 1, which will lean heavily on the privacy guarantee given in Theorem 12. We note that if our mechanism were $(\epsilon, 0)$-differentially private then a result of [33], stating that any $(\epsilon, 0)$-DP mechanism is $2\epsilon$-dominant strategy incentive compatible.

Two issues arise if one were to try this approach in our setting. First, the entire mechanism is not differentially private as discussed above. A bidder $i$'s behavior might have significant impact on other bidder's allocations and payments, and those bidders may as a result choose to behave differently in later rounds based on that information. Thus we relax to the weaker incentive guarantee of utility-approximate BIC, avoiding the issue of other bidders behaving differently in response to activity from earlier rounds. Second, the stream of estimates maintained by our mechanism is $(\epsilon, \delta)$-differentially private for $\delta = \epsilon/T > 0$ and not $(\epsilon, 0)$-differentially private which is necessary for the result of [33] to hold.

**Theorem 13.** *Algorithm 1 is $kh\epsilon\left(2 + \frac{1}{T}\right)$-utility approximate BIC when $\epsilon < 1$.*

Finally, we consider the revenue-optimality of our proposed mechanism. Our revenue guarantees will rely on tools developed and described in Appendix A.2. Recall that we use $\text{Rev}(\mathcal{M}; D)$ to denote the expected revenue generated by the mechanism $\mathcal{M}$ on a value distribution $D$, and that $\mathbf{D}$ and $\mathbf{D}'$ respectively denote the joint distributions of true values and true values rounded down to the nearest multiple of $\beta$. Let $\mathcal{M}^*_{\mathbf{H}'_t}$, $\mathcal{M}^*_{\mathbf{D}'}$, and $\mathcal{M}^*_{\mathbf{D}}$ be the truly revenue-optimal mechanisms for the distributions $\mathbf{H}'_t$, $\mathbf{D}'$, and $\mathbf{D}$, respectively. In each round of our mechanism, we get a sample from $\mathbf{D}'$ and run Myerson's auction with $\mathbf{H}'_t$ as the prior; that is, we run $\mathcal{M}^*_{\mathbf{H}'_t}$. We use $\text{Rev}(\mathcal{M}^*_{\mathbf{H}'_t}; \mathbf{D}')$ to denote the expected revenue we achieve in round $t$. The bidders true values are drawn from $\mathbf{D}$ and we use $\text{Rev}(\mathcal{M}^*_{\mathbf{D}}; \mathbf{D})$ to denote the optimal expected revenue in any round. In this section, we will compare the revenue $\text{Rev}(\mathcal{M}^*_{\mathbf{H}'_t}; \mathbf{D}')$ of our mechanism to the optimal revenue $\text{Rev}(\mathcal{M}^*_{\mathbf{D}}; \mathbf{D})$.

The main result of this section, Theorem 14, bounds the difference between our average expected revenue, $\frac{1}{T}\sum_{t=1}^{T} \text{Rev}(\mathcal{M}^*_{\mathbf{H}'_t}; \mathbf{D}')$, and that of the optimum. We show that over $T$ rounds, with high probability the average expected revenue is within an additive $o(1)$ error of the optimal.

**Theorem 14.** *With probability at least $1 - \alpha$, Algorithm 1 satisfies $\frac{1}{T}\sum_{t=1}^{T} \text{Rev}(\mathcal{M}^*_{\mathbf{H}'_t}; \mathbf{D}') \geq \text{Rev}(\mathcal{M}^*_{\mathbf{D}}; \mathbf{D}) - \beta J - 4hn^2 \tilde{O}\left(\frac{1}{\sqrt{T}} + \frac{1}{T\epsilon}\right)$ for regular distributions $\mathbf{D}$ and $\epsilon < 1$.*

*Proof.* We start by instantiating Lemma 34 for every round $t$ instantiated with failure probability $\alpha/T$. Then taking a union bound over all $T$ rounds and summing over $t$, ensures that with probability $1 - \alpha$, $\frac{1}{T}\sum_{t=1}^{T} \text{Rev}(\mathcal{M}^*_{\mathbf{H}'_t}; \mathbf{D}') \geq \text{Rev}(\mathcal{M}^*_{\mathbf{D}}; \mathbf{D}) - \beta J - \frac{4hn^2}{T}\sum_{t=1}^{T}\gamma_t$ for

$$\gamma_t = \sqrt{\frac{\log\frac{n}{\alpha}}{2t}} + \frac{\sigma}{t}\sqrt{\log\frac{h}{\beta}\log T}\sqrt{2\log\left(\frac{2hn}{\beta\alpha}\right)} \text{ and } \sigma = \frac{8\log T \log\frac{h}{\beta}}{\epsilon}\sqrt{\ln\frac{T\log T\log\frac{h}{\beta}}{\epsilon}}$$

In the remainder of the proof, we bound $\frac{1}{T}\sum_{t=1}^{T}\gamma_t$. (Recall that the $\alpha$ in Lemma 34 is $\alpha/T$ here.)

$$\frac{1}{T}\sum_{t=1}^{T}\gamma_t = \frac{1}{T}\sum_{t=1}^{T}\left(\sqrt{\frac{\log\frac{nT}{\alpha}}{2t}} + \frac{\sigma}{t}\sqrt{\log\frac{h}{\beta}\log T}\sqrt{2\log\left(\frac{2hnT}{\beta\alpha}\right)}\right)$$

$$\leq \sqrt{\frac{2\log\frac{nT}{\alpha}}{T}} + \frac{2\sigma\log T}{T}\sqrt{\log\frac{h}{\beta}\log T}\sqrt{2\log\left(\frac{2hnT}{\beta\alpha}\right)} = \tilde{O}\left(\frac{1}{\sqrt{T}} + \frac{1}{T\epsilon}\right)$$

The first inequality comes from the facts that $\sum_{t=1}^{T}\frac{1}{\sqrt{t}} \leq 2\sqrt{T}$ and $\sum_{t=1}^{T}\frac{1}{t} = H_T \leq \log T + 1 \leq 2\log T$. The following equality come from plugging in the expression of $\sigma$ and combining terms.

Hence, w.p. $\geq 1 - \alpha$, $\frac{1}{T}\sum_{t=1}^{T} \text{Rev}(\mathcal{M}^*_{\mathbf{H}'_t}; \mathbf{D}') \geq \text{Rev}(\mathcal{M}^*_{\mathbf{D}}; \mathbf{D}) - \beta J - 4hn^2\tilde{O}\left(\frac{1}{\sqrt{T}} + \frac{1}{T\epsilon}\right)$. $\square$

Notice that this statement says that if we set $\beta$, the discretization parameter to be $o(1)$ in terms of $T$ then the revenue one can earn is a $(1 - o(1))$-approximation to the optimal revenue.

# 4 Bid-Approximate Bayesian Incentive Compatibility

We now describe an algorithm for "training" a mechanism to achieve nearly optimal revenue in an exact equilibrium, where each bidder bids close to their true value. This contrasts with the result in the prior section, wich shows that bidders bidding their exact true values is an approximate equilibrium. While the equilibrium we describe in this section is not quite a truthful one, we can compare its revenue with the revenue of the optimal mechanism facing truthful bids.

The primary technical meat of this section has two parts. First, we describe how to modify Algorithm 1 to get the stronger incentive guarantee of bid-approximate Bayesian incentive compatibility. Theorem 13 gave utility-approximate BIC, which means all bidders behaving truthfully in all rounds is an approximate Bayes-Nash equilibrium (BNE). Bid-approximate BIC means there is an *exact* Bayes-Nash equilibrium where all bidders bid within $\eta$ of their values. The main challenge here comes from whether bidders change their behavior in later rounds based on the non-truthful behavior of other bidders in earlier rounds.[2] To make this jump, we guarantee that round $t$'s allocations and payments leak very little information about a bidder's behavior to other bidders. So, bidders will have very little ability to condition on one another's behavior in round $t$ when selecting a strategy in later rounds. To this end, we ensure that the allocations in round $t$ are differentially private and the payments in the round $t$ are jointly differentially private, informally stated below.

**Theorem 15.** *Algorithm 2 runs in polynomial time, is $\eta_t$-bid approximate BIC in round $t$ for $\eta_t = c + \tilde{\mathcal{O}}(t^{-1/4})$ and achieves expected revenue $Rev = OPT - c' - \tilde{\mathcal{O}}(t^{-1/4})$ for small constants $c, c'$.*

The more precise result, given in Theorem 19, shows that under mild assumptions about bidders' behavior, every equilibrium bid under our mechanism lies close to the true value in the bid space. That is, in round $t$, we show that no bidder in equilibrium will underbid by more than a small factor $\eta_t$. The main challenge in getting such a result requires us to bound the extent to which bidders' behavior in the current round can affect their future utilities. If we can control this quantity, we can control the amount by which people can game the system in the current round and also ensures that bidders are unable to learn much about the value distributions of other bidders. The formal revenue guarantees for our mechanism are are presented in Theorem 20 where we show that with high probability, the average expected revenue obtained by our mechanism is close to the optimal expected revenue of the mechanism with complete information about the value distribution of the bidders (i.e., Myerson's auction on $\mathbf{D}$).

## 4.1 A mechanism with private payments and allocations

We now Algorithm 2 which is a Bid-approximate BIC Online Auction algorithm. The primary challenge is to ensure that the allocation and payments in round $t$ do not leak much information about bidders' behavior to one another. The mechanism ensures this by making choices which are all jointly differentially private, ensuring that bidders do not substantially affect either the mechanism's state or what other bidders learn about them. The mechanism maintains a private estimate of the empirical distribution (as before), and also computes prices and allocations in round $t$ using jointly differentially private algorithms. We use the exponential mechanism (Algorithm 6) for picking round $t$'s allocation, black box payments (Algorithm 5) to pick payments in round $t$ (which in expectation yields payments close to the truthful payments). This additional step of ensuring round $t$ decisions are differentially private is crucial when bidders might condition their behavior in later rounds based upon what the learn from round $t$ (which might be the case in bid-approximate equilibria).

To describe exact equilibria, we need to argue about each bidder's utility for modifying her bid, and argue that shading by more than $\eta_t$ hurts her utility. We do this using a *punishing* mechanism, which penalizes bidders who shade their bid by using a strictly truthful mechanism with some probability $\rho_t$ in each round. Since we are going to show that there exists an equilibrium where in round $t$ every bidder bids with $\eta_t$ of their true value, it will be convenient to define the round $t$ equilibrium bid distributions $\mathbf{F}_t = \mathbf{F}_{1.t} \times \cdots \times \mathbf{F}_{n.t}$ .

**Algorithm 2:** Bid-approximate BIC Online Auction

---
**Parameters:** discretization $\beta$, privacy $\epsilon$, upper bound on support $h$, num. of rounds $T$
**Initialize:** $H'_{i,0} \leftarrow \text{Uniform}(0, h)$ for $i = 1, \cdots, n$
**for** $t = 1, \cdots, T$ **do**
    Receive vector of bids $\mathbf{b}_t = (b_{1,t}, \ldots, b_{n,t})$, rounded down to multiple of $\beta$
    With probability $\rho_t$ run mechanism StrictlyTruthful($\mathbf{b}_t$) (Algorithm 4)
    **else**
        **for** $i = 1, \ldots, n$ **do**
            Use $H'_{i,t-1}$ to calculate $\phi_{i,t}(b_{i,t})$ (Theorem 3)
        **end**
        Use exponential mechanism (Algorithm 6) to select allocation $x_t(\phi_{\mathbf{t}}(\mathbf{b}))$
        Use *Black box payments* (Algorithm 5) to calculate payments $p_t(\mathbf{b}_t)$.
    **end**
    **for** $i = 1, \ldots, n$ **do**
        Update $H'_{i,t}$ via two-fold tree aggregation (Algorithm 3), giving as input $b_{i,t}$
    **end**
**end**

---

## 4.2 Privacy, Truthfulness, and Revenue Guarantees

In this subsection, we provide the differential privacy and incentive guarantees of Algorithm 2. We refer to Appendix B for the and additional technichal details and proofs.

**Lemma 16.** *Algorithm 2 is $(3\epsilon, 3\epsilon/T)$-jointly differentially private in the bids of users.*

**Lemma 17.** *Fix a round $t$, population $i$, and the bidder from population $i$ in round $t$. This bidder's total utility in rounds $t' > t$ for misreporting in round $t$ is $\leq \epsilon k h(2 + \frac{1}{T})$ more than truthful reporting.*

Finally, we introduce an assumption we make about the equilibrium behavior of bidders facing this mechanism. Informally, the assumption states that a mechanism run over many rounds should have equilibrium bid distributions which have similar probability of any bid between adjacent rounds.

**Assumption 18** ($\boldsymbol{\lambda}$-Stable Bid Distribution)**.** *We say the mechanism $\mathcal{M}$ supports a $\boldsymbol{\lambda}$-stable bid distribution if it has some BNE with distribution of equilibrium bids $\mathbf{F}_t$ in round $t$ such that for all populations $i$ and rounds $t$, there exist $\lambda_t$ s.t. $\|\mathbf{F}_t - \mathbf{F}_{t-1}\|_\infty \leq \lambda_t$.*

**Remark.** Consider a mechanism with very similar behavior in round $t$ versus $t + 1$: both the mechanism's distribution over allocation and payment rules in round $t$ and $t+1$ are very close. If all other bidders strategies from round $t$ to $t + 1$ are also very close, then the utility for any particular bid $b$ from a bidder with valuation $v$ from population $i$ will be quite close, but that bidder's utility-optimal bid may or may not be identical or even particularly close in the two rounds.

This suggests that analyzing exact equilibria in iterated settings is quite complex, in that the distribution over utility-optimal bids might shift quite substantially from round to round. So, mechanisms and equilibria without this property may have highly erratic behavior, and such equilibria may not support a learning procedure which competes with the (truthful) optimal revenue. Hence we assume that the mechanism in Algorithm 2 supports a $\boldsymbol{\lambda}$-Stable Bid Distribution with the condition that the quantity $\Sigma_T := \sum_{t=1}^{T-1} \frac{t\lambda_t}{T} = o(1)$. We now present the truthfulness gurantee of Algorithm 2.

**Theorem 19.** *Algorithm 2 in round $t$ is $\eta_t$-bid approximate BIC, i.e. in round $t$, any bidder $i$ with value $v_{i,t}$ reports bid $b_{i,t}$ which satisfies $v_{i,t} - \eta_t \leq b_{i,t} \leq v_{i,t}$ where $\eta_t = h\sqrt{\frac{8n^2\gamma_t + 6k\epsilon}{\rho_t J}}$,*

$$\gamma_t = \sqrt{\frac{\log(2hn/\beta\alpha)}{2t}} + \Sigma_t + \frac{\sigma}{t}\sqrt{\log\frac{h}{\beta}\log T}\sqrt{\log\left(\frac{hn}{\beta\alpha}\right)}, \ \sigma = \frac{8\log T\log\frac{h}{\beta}}{\epsilon}\sqrt{\ln\frac{\log T\log\frac{h}{\beta}}{\delta}} \ and \ \delta = \frac{\epsilon}{T}.$$

Let $\mathbf{F}'_t$ be distribution of round $t$ bids ($\mathbf{F}_t$) rounded down to the nearest multiple of $\beta$. Let the mechanism we run in round $t$ on the bids we recieve from $\mathbf{F}'_t$ be $\mathcal{M}_{\mathbf{H}'_t}$. The expected revenue we achieve in round $t$ is $\text{Rev}(\mathcal{M}_{\mathbf{H}'_t}; \mathbf{F}'_t)$ which we will compare with the optimal reveue $\text{Rev}(\mathcal{M}^*_{\mathbf{D}}; \mathbf{D})$. We now present the main revenue guarantee for Algorithm 2, i.e we compare the average expected revenue of Algorithm 2 with respect to optimal revenue. We refer to the Appendix B.4 for the proof.

**Theorem 20.** *Using Algorithm 2 for $T$ rounds, with probability at least $1 - \alpha$ the average expected revenue obtained by the mechanism over the $T$ rounds satisfies,*

$$\frac{1}{T}\sum_{t=1}^{T} Rev(\mathcal{M}_{\mathbf{H'}_t}; \mathbf{F}'_t) \geq Rev(\mathcal{M}_{\mathbf{D}}^*; \mathbf{D})$$

$$- \left( hnJ^{2/3}\tilde{\mathcal{O}}(\tfrac{1}{T^{1/4}}) + \tfrac{J \ln n}{\epsilon} + \tfrac{h}{3\epsilon}\ln\tfrac{J}{k\epsilon} + hJ^{2/3}(12k\epsilon)^{1/3} + \beta J \right)$$

## Footnotes

[1]Our results can be shown to hold when values are not drawn fresh, in which case $D_i$ is the distribution of drawn values, taking into account the process by which bidders are redrawn from population.

[2]This is not a challenge when we assume all bidders behave truthfully, since bidders won't have non-truthful behavior in earlier rounds to condition their behavior upon.

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
