[Supplementary Material]

# A Omitted Proofs and Subroutines for Utility-Approximate BIC Online Auction (Algorithm 1)

## A.1 Differentially Private Distribution Estimation

We now describe a differentially private procedure for estimating value distributions $H'_{i,t}$ for each bidder population. This corresponds to the final operation in each round of our mechanism (Algorithm 1).

When we receive value $v_{i,t}$ we round it down to nearest integer multiple of $\beta$. Recall that the value distribution for bidder population $i$ rounded down to the nearest multiple of is $D'_i$, and that $D'_i$ has finite support $\{0, \beta, 2\beta, \cdots, h\}$ of size $h/\beta + 1$, and thus the estimates $\mathbf{H}'_{-i,t}$ we maintain will have the same finite support. We work under the assumption that bids are truthfully reported values, which we later validate with the incentive guarantees shortly.

Let $H_{i,t}$ be the empirical (non-private) estimate of $D'_i$ at round $t$. The following lemma establishes that the empirical distribution of rounded values provides a good estimate of the true distribution of rounded values, with respect to the supremum norm $\|\cdot\|_\infty$.

**Lemma 21.** *Let $H_{i,t}$ be the empirical distribution of $t$ i.i.d. samples from $D'_i$. Then, with probability at least $1 - \alpha$, $\|D'_i - H_{i,t}\|_\infty \leq \sqrt{\frac{\log \frac{2}{\alpha}}{2t}}$.*

*Proof.* This is a direct result of the Dvoretzky-Kiefer-Wolfowitz inequality [32], which establishes the concentration of the empirical CDF for any distribution. $\qquad\square$

### A.1.1 Two-fold tree aggregation

Standard tree-based aggregation [20, 11] maintains online counts of a single quantity (e.g., the number of 1s in a stream of bits); we will use an extension of these known as *two-fold tree aggregation* [30] to maintain an online estimate of the CDF of a probability distribution $\mathbf{D}'$ in a differentially private manner. Informally, one achieves this by maintaining a private counter for each bin in a histogram over possible values which might update the counters. Since $\mathbf{D}'$ has a discrete support of $\{0, \beta, 2\beta, \cdots, h\}$, the non-private empirical CDF $H_{i,t}$ can be described by a simple (increasing) step function, with steps occurring at integer multiples of $\beta$. To compute $H_{i,t}(u)$, the empirical CDF at value $u$, we need only count the number of samples from $v_{i,1}, \cdots, v_{i,t}$ which are less than $u$, i.e., $H_{i,t}(u) = (\sum_{\tau=1}^{t} \mathbf{1}\{v_{i,\tau} \leq u\})/t$. Two-fold tree aggregation allows us to privately maintain these cumulative sums for all points $u \in \{\beta, 2\beta, \cdots, h\}$ in the support of our distributions.

The algorithm maintains $n$ instances of the two-fold tree aggregation procedure, one for each bidder population, where each instance has its own distinct internal state. The $i$th instance maintains $H'_{i,t}$ in round $t$. In each round $t$, the mechanism receives a value profile $\mathbf{v}_t$. For each population $i$, $v_{i,t}$ is used to update the internal state of population $i$'s tree aggregation instance.

The algorithm is given formally in Algorithm 3, and require the following additional notation. Consider any $t \in [T]$ with binary representation $(t_{\lceil \log T \rceil}, \ldots, t_1, t_0)$. That is, $t = \sum_0^{\lceil \log T \rceil} t_j 2^j$. Let $j_t$ be the lowest nonzero bit, and let

$$\Lambda_t = \left\{ t - 2^{j_t} + 1, t - 2^{j_t} + 2, \cdots, t - 1, t \right\}.$$

We also define the set,

$$\Gamma_t = \{t' : t' = t - \sum_{j=0}^{h-1} t_j 2^h, h = 1, \ldots, \lceil \log T \rceil\}.$$

We note that $\Gamma_t$ has size at most $\lceil \log T \rceil$, and the set $[t]$ can be described as the union of $\lceil \log T \rceil$ sets, i.e., $[t] = \cup_{j \in \Gamma_t} \Lambda_j$.

In two-fold tree aggregation, we have these aggregations over two axes: time $t$ and value $u$. Thus we maintain $\frac{h}{\beta} \cdot T$ partial sums, denoted in Algorithm 3 as internal states $A_{tq} = H_{i,t}(u)$ for $u = q\beta$. One sample $v_{i,t}$ contributes to at most $\log \frac{h}{\beta} \log T$ partial sums, and each $H'_{i,t}(u)$ can be written as a sum of at most $\log \frac{h}{\beta} \log T$ partial sums.

---

**Algorithm 3:** Two-fold tree aggregation for population $i$ [30]

---

**Input:** discretization parameter $\beta$, privacy parameter $\epsilon$, upper bound on support $h$, number of rounds $T$

**Internal State:** Noisy partial sums $A_{tq}$ for all $t \in [T]$ and $q \in \left[\frac{h}{\beta}\right]$

**Initialize:** Set $\sigma = \frac{8 \log T \log \frac{h}{\beta}}{\epsilon} \sqrt{\ln \frac{T \log T \log \frac{h}{\beta}}{\epsilon}}$ and sample $A_{tq} \sim_{i.i.d.} \mathcal{N}(0, \sigma^2)$ for all $t$ and $q$

**for** $t = 1, \cdots, T$ **do**

    Receive $v_{i,t} = p\beta$ for some $p \in \left[\frac{h}{\beta}\right]$

    **for** $j, k$ satisfying $t \in \Lambda_j$ and $p \in \Lambda_k$ **do**

    |  $A_{jk} = A_{jk} + 1$

    **end**

    **for** $q \in \left[\frac{h}{\beta}\right]$ **do**

    |  Sample $\nu_{tq} \sim \mathcal{N}(0, ((\log \frac{h}{\beta} + 1)(\log T + 1) - |\Gamma_t||\Gamma_q|)\sigma^2)$

    **end**

    Output $H'_{i,t}$, the estimated CDF:    $H'_{i,t}(x) := \sum_{j \in \Gamma_t} \sum_{k \in \Gamma_q} \frac{A_{jk} + \nu_{tq}}{t}$,    where $q = \lfloor x/\beta \rfloor$.

**end**

---

The following lemma shows that Algorithm 3 is differentially private, and guarantees that privacy is maintained throughout the entire run of the algorithm.

**Lemma 22** (Liu et al. [30]). *The entire stream of estimates $\{H'_{i,t}\}_{t=1}^T$ output by Algorithm 3 is $\left(\epsilon, \frac{\epsilon}{T}\right)$-differentially private with respect to the input stream of bids $\{v_{i,t}\}_{t=1}^T$.*

The construction of Algorithm 3 ensures that every value $H'_{i,t}(u)$ is obtained by perturbing $H_{i,t}(u)$ with the $t$-normalized sum of Gaussian variables, each with variance $\sigma^2$. The total number of Gaussian noise terms added to obtain $H'_{i,t}(u)$ is no more than $\log \frac{h}{\beta} \log T$ because the sets $\Gamma_t$ and $\Gamma_q$ used in the final output of Algorithm 3 have size at most $\lceil \log T \rceil$ and $\lceil \log h/\beta \rceil$, respectively. That is, for each fixed $u$, we have $H'_{i,t}(u) - H_{i,t}(u) \sim N\left(0, \frac{\sigma^2}{t^2} \log \frac{h}{\beta} \log T\right)$. Lemma 23 uses this fact to bound the distance between $H_{i,t}$ and $H'_{i,t}$.

**Lemma 23.** *After $t$ rounds, for a fixed population $i$, with probability at least $1 - \alpha$ the empirical distribution $H_{i,t}$ and the differentially private estimate $H'_{i,t}$ produced by Algorithm 3 will satisfy*

$$\left\| H_{i,t} - H'_{i,t} \right\|_\infty \leq \frac{\sigma}{t} \sqrt{\log \frac{h}{\beta} \log T} \sqrt{2 \log\left(\frac{2h}{\beta\alpha}\right)},$$

*for $\sigma = \frac{8 \log T \log \frac{h}{\beta}}{\epsilon} \sqrt{\ln \frac{T \log T \log \frac{h}{\beta}}{\epsilon}}$.*

*Proof.* For a Gaussian random variable $Z \sim \mathcal{N}(\mu, \rho^2)$ it holds that $\mathrm{P}[|Z - \mu| > x\rho] \leq 2\exp(-x^2/2)$, by a standard tail bound. We apply this inequality to each point $u \in \{0, \beta, 2\beta, \cdots, h\}$ to see that, with probability at least $1 - \frac{\beta\alpha}{h}$, we have $\left| H_{i,t}(u) - H'_{i,t}(u) \right| \leq \frac{\sigma}{t} \sqrt{\log \frac{h}{\beta} \log T} \sqrt{2 \log\left(\frac{2h}{\beta\alpha}\right)}$. Applying a union bound over all of the $\frac{h}{\beta}$ values of $u$ completes the proof. $\square$

We can now combine the previous lemmas to relate the distributions $H'_{i,t}$ and $D'_i$.

**Theorem 24.** *After $t$ rounds Algorithm 1, it holds with probability at least $1 - \alpha$ that*

$$\left\| H'_{i,t} - D'_i \right\|_\infty \leq \gamma_t \quad \text{for every } i \in [n],$$

*where $\gamma_t = \sqrt{\frac{\log \frac{n}{\alpha}}{2t}} + \frac{\sigma}{t} \sqrt{\log \frac{h}{\beta} \log T} \sqrt{2 \log\left(\frac{2hn}{\beta\alpha}\right)}$ and $\sigma = \frac{8 \log T \log \frac{h}{\beta}}{\epsilon} \sqrt{\ln \frac{T \log T \log \frac{h}{\beta}}{\epsilon}}$.*

*Proof.* Applying triangle inequality on Lemma 23 and Lemma 21 gives the bound for a single bidder population $i$, and a union bound over all $n$ bidder populations proves the lemma. □

In the remainder of this section, the same definition of $\gamma_t$ will be used.

## A.2 Revenue Maximization on Similar Distributions

In the previous section we showed that the differentially private estimate of the value distribution we maintain is close to the true distribution from where the values are being drawn. We now introduce tools we need to argue that the revenue of our mechanism, facing truthful bidding, is approximately optimal. In particular, describe how one can compare the revenue of a fixed, well-behaved mechanism on two similar product distributions $\mathbf{D}$ and $\tilde{\mathbf{D}}$. This will imply that the problem of optimizing with respect to $\tilde{\mathbf{D}}$ will yield approximately optimal revenue with respect to $\mathbf{D}$. These results are broadly stated and should be of independent interest.

More formally, we will consider distributions that are close in $\ell_\infty$ distance, and mechanisms which are well-behaved in the sense that allocating one agent leads to the exclusion of others. The relevant definitions follow. Recall that we overload the notation and $D$ and $\tilde{D}$ refer to the corresponding CDFs when used as functions.

**Definition 25** ($\tau$-closeness). *We call two distributions $D$ and $\tilde{D}$ $\tau$-close if $\|D - \tilde{D}\|_\infty \le \tau$.*

**Definition 26** (Competitiveness). *A truthful mechanism is* competitive *if for any valuation profile $\mathbf{v}$ and any pair of bidders $i$ and $j$, the allocation probability $x_i(\mathbf{v})$ for bidder $i$ is a non-increasing function of $v_j$.*

In multi-unit settings, mechanisms that exactly maximize virtual surplus for any monotone virtual value function (e.g. ironed Myerson virtual value) satisfy this property, as does the exponential mechanism with score for each allocation equal to its virtual surplus. Given these definitions, we may state the main result of this section:

**Theorem 27.** *Let $\mathcal{M}$ be a competitive mechanism, and let $\mathbf{D}$ and $\tilde{\mathbf{D}}$ be two product distributions of values such that for every bidder $i$, $D_i$ and $\tilde{D}_i$ are $\tau$-close. Then the expected revenue of $\mathcal{M}$ on $\mathbf{D}$ is within an additive $2n^2h\tau$ of the revenue from $\mathcal{M}$ on $\tilde{\mathbf{D}}$. That is, $|Rev(\mathcal{M};\mathbf{D}) - Rev(\mathcal{M};\tilde{\mathbf{D}})| \le 2n^2h\tau$.*

In fact, we prove a stronger statement than Theorem 27: we show that the revenue from each bidder is within $2nh\tau$ in each mechanism. To prove this stronger statement, we argue from the perspective an individual bidder, and consider three steps. First, we show that switching the distributions of all other bidders from $\mathbf{D}_{-i}$ to $\tilde{\mathbf{D}}_{-i}$ does not significantly change $i$'s allocation probability. We then show that because of the relationship between allocation and payments in truthful mechanisms an insignificant change in allocation probability implies an insignificant change in revenue. We then show that switching bidder $i$'s distribution from $D_i$ to $\tilde{D}_i$ does not significantly impact the revenue of any mechanism. The result will follow from the triangle inequality. We begin with the first of our three steps by defining new notation for the allocation probability secured by each bidder in expectation over the other bidders.

**Definition 28** (Interim allocation rule). *Let $\mathcal{M} = (\mathbf{x}, \mathbf{p})$ be a mechanism for the single-round game. For any value $v_i$, the* interim allocation rule *for $i$ at $v_i$ is given by $x_i(v_i) = \mathbb{E}_{\mathbf{v}_{-i}}[x_i(\mathbf{v})]$.*

Our first step is to show that each bidder's interim allocation rule under competitive mechanisms is robust to small changes in other bidders' value distributions. Formally:

**Lemma 29.** *Let $\mathbf{D}_{-i}$ and $\tilde{\mathbf{D}}_{-i}$ be value distributions for bidders other than $i$, with $D_j$ and $\tilde{D}_j$ $\tau$-close for all $j \ne i$. Consider any truthful competitive mechanism, and let $x_i(\cdot)$ and $\tilde{x}_i(\cdot)$ denote the interim allocation rules of bidder $i$ under $\mathbf{D}_{-i}$ and $\tilde{\mathbf{D}}_{-i}$, respectively. Then for any value $v_i$, $|x_i(v_i) - \tilde{x}_i(v_i)| \le (n-1)\tau$.*

*Proof.* We will consider changing the value of just one bidder, $j$, and observing the impact on the interim allocation rule of bidder $i$. The lemma will follow from repeating this argument once for

each bidder other than $i$. To show that slightly changing bidder $j$'s distribution has a minimal effect, we write the following sequence of equalities and inequalities, which we justify afterward.

$$x_i(v_i) = \int_0^h \mathbb{E}_{\mathbf{v}_{-i,j}}[x_i(\mathbf{v})]D_j(v_j)\,dv_j$$

$$= \mathbb{E}_{\mathbf{v}_{-i,j}}[x_i(\mathbf{v})]D_j(v_j)\Big|_0^h - \int_0^h \frac{d}{dv_j}\mathbb{E}_{\mathbf{v}_{-i,j}}[x_i(\mathbf{v})]D_j(v_j)\,dv_j$$

$$= \mathbb{E}_{\mathbf{v}_{-i,j}}[x_i(v_i, h, \mathbf{v}_{-i,j})] - \int_0^h \mathbb{E}_{\mathbf{v}_{-i,j}}[\tfrac{d}{dv_j}x_i(\mathbf{v})]D_j(v_j)\,dv_j$$

$$\geq \mathbb{E}_{\mathbf{v}_{-i,j}}[x_i(v_i, h, \mathbf{v}_{-i,j})] - \int_0^h \mathbb{E}_{\mathbf{v}_{-i,j}}[\tfrac{d}{dv_j}x_i(\mathbf{v})](\tilde{D}_j(v_j) - \tau)\,dv_j$$

$$= \mathbb{E}_{\mathbf{v}_{-i,j}}[x_i(v_i, h, \mathbf{v}_{-i,j})] - \int_0^h \mathbb{E}_{\mathbf{v}_{-i,j}}[\tfrac{d}{dv_j}x_i(\mathbf{v})]\tilde{D}_j(v_j)\,dv_j - \tau$$

$$= \mathbb{E}_{\mathbf{v}_{-i,j}}[x_i(\mathbf{v})]\tilde{F}_j(v_j)\Big|_0^h - \int_0^h \mathbb{E}_{\mathbf{v}_{-i,j}}[\tfrac{d}{dv_j}x_i(\mathbf{v})]\tilde{D}_j(v_j)\,dv_j - \tau$$

$$= \int_0^h \mathbb{E}_{\mathbf{v}_{-i,j}}[x_i(\mathbf{v})]\tilde{D}_j'(v_j)\,dv_j - \tau$$

The reasoning is as follows. The first equality is from the definition of the interim allocation rule $x_i(v_i)$. The second and third equalities follow by integration by parts and interchanging the derivative and integral, respectively. The third inequality follows from the $\tau$-closeness of $D_j$ and $\tilde{D}_j$, and from the fact that $\frac{d}{dv_j}x_i(\mathbf{v}) \leq 0$ by the competitiveness of the mechanism. The remaining equalities follow from the same reasoning as the first three. Hence, changing bidder $j$'s value distribution from $D_j$ to $\tilde{D}_j$ can decrease bidder $i$'s allocation probability by at most $\tau$. A symmetric argument bounds the increase. Further applying this same argument to all bidders other than $i$, one at a time, implies the lemma. □

The payment identity (Theorem 2) characterizes the payments of an individual bidder with a realized type in a truthful mechanism, and shows that this payment is completely determined by the allocation rule the agent faces. Taking expectations over the values of other agents yields a characterization of an agent's expected payments in terms of their interim allocation rule. This characterization will allow us to show that the revenue from any bidder under two similar interim allocation rules is similar.

**Corollary 29.1** (of Theorem 2). *In any truthful mechanism, for any bidder $i$ with value $v_i$, the expected revenue of bidder $i$ satisfies:*

$$\mathbb{E}_{\mathbf{v}_{-i}}[p_i(\mathbf{v})] = v_i x_i(v_i) - \int_0^{v_i} x_i(z)\,dz + \mathbb{E}_{\mathbf{v}_{-i}}[p_i(0, \mathbf{v}_{-i})] \tag{1}$$

**Lemma 30.** *Let $x_i$ and $\tilde{x}_i$ be interim allocation rules for bidder $i$ such that $|x_i(v_i) - \tilde{x}_i(v_i)| \leq \tau$ for all $v_i \in [0, h]$. If $\mathbb{E}_{\mathbf{v}_{-i}}[p_i(0, \mathbf{v}_{-i})] = 0$ under both allocation rules, then for any value $v_i$, the expected payments made by a bidder with that value differ by at most $2v_i\tau$ under the two allocation rules.*

*Proof.* By equation (1), the difference in revenue between the two mechanisms is given by

$$v_i(x_i(v_i) - \tilde{x}_i(v_i)) - \int_0^{v_i}(x_i(z) - \tilde{x}_i(z))\,dz.$$

The first term is at most $v_i\tau$. Moreover the second term is at most $\int_0^{v_i} \tau\,dz$, which is equal to $v_i\tau$. □

Combining Lemmas 29 and 30 yields:

**Corollary 30.1.** *Let $\mathbf{D}_{-i}$ and $\tilde{\mathbf{D}}_{-i}$ be value distributions for bidders other than $i$, with $D_j$ and $\tilde{D}_j$ $\tau$-close for all $j \neq i$. Consider any truthful competitive mechanism $\mathcal{M}$ where the bidders with value $0$ make no payments. Then the expected revenue of $\mathcal{M}$ from bidder $i$ differs by at most $2(n-1)h\tau$.*

We finally show that holding other bidders' value distributions fixed and switching bidder $i$ from value distribution $D_i$ to a $\tau$-close distribution $\tilde{D}_i$ yields similar revenue. Formally:

**Lemma 31.** *Let $D_i$ and $\tilde{D}_i$ be $\tau$-close value distributions for bidder $i$. For any truthful mechanism $\mathcal{M}$ and any value distributions $\mathbf{D}_{-i}$ for other bidders, the expected revenue from bidder $i$ under $D_i \times \mathbf{D}_{-i}$ and $\tilde{D}_i \times \mathbf{D}_{-i}$ differ by at most $h\tau$.*

To prove Lemma 31, we use a standard characterization of a bidder's expected payments in a truthful mechanism, which can be obtained by integrating equation (1) over all values $v_i$ and integrating by parts.

**Corollary 31.1** (of Theorem 2)**.** *In any truthful mechanism where bidders with value $0$ make no payments, for any bidder $i$ with value distribution $D_i$, the expected revenue from bidder $i$ is given by*

$$\mathbb{E}_{\mathbf{v}}[p_i(\mathbf{v})] = \int_0^h x_i'(v_i) R_i(v_i) \, dv_i \tag{2}$$

*where $R_i(v_i)$ is bidder $i$'s price posting revenue function, given by $R_i(v_i) = v_i(1 - D_i(v_i))$*

*Proof of Lemma 31.* By equation (2), the difference in expected revenue between the two distributions is given by

$$\int_0^h x_i'(v_i) v_i (\tilde{D}_i - D_i) \, dv_i \leq h\tau \int_0^h x_i'(v_i) \, dv_i$$

Since $\int_0^h x_i'(v_i) \, dv_i \leq 1$, the result follows. $\qquad\square$

*Proof of Theorem 27.* Combining Corollary 30.1 with Lemma 31 and using the triangle inequality implies that the revenue of any individual bidder $i$ differs by at most $2(n-1)h\tau + h\tau \leq 2nh\tau$ under $\mathbf{D}$ rather than $\tilde{\mathbf{D}}$. Summing over all bidders yields the desired bound. $\qquad\square$

## A.3  Omitted Proofs in Section 3

*Proof of Theorem 12.* Algorithm 1's only record of bids which persists across rounds is its distribution estimate $\mathbf{H}'_t$. In each round, it chooses an auction as a post-processing step over those estimated distributions. The two-fold tree aggregation step is $(\epsilon, \epsilon/T)$-differentially private by Lemma 22. Thus, the algorithm's post-processing to estimate the virtual value distribution and select future auctions is $(\epsilon, \epsilon/T)$-differentially private by Lemma 9. $\qquad\square$

*Proof of Theorem 13.* Consider a bidder deciding how to bid in round $t$. She has three considerations: how her behavior will affect (1) the learning algorithm in future rounds, (2) the behavior of other bidders in future rounds, and (3) her utility in round $t$.

Since we seek to show that the mechanism is utility-approximate BIC, we can assume all other bidders behave truthfully in every history (by Definition 6). Therefore, other bidders will not change their behavior in future rounds, and the value of (2) is 0. The value of (3) is also 0 because the empirical Myerson auction run in each round $t$ is chosen to be exactly truthful as a one-shot (static) mechanism, so no payer can gain in expected utility by misreporting her bid. Thus the only utility a player can gain by lying about her value is from (1).

Next we analyze (1). Since our mechanism's differential privacy guarantee limits the extent to which a player $i$'s report in round $t$ affects $H'_{i,s}$ for each $s > t$, and hence limits how it affects future choices of the mechanism, this allows us to control the amount of future utility she can gain from misreporting at $t$. Consider the change in this player's utility in all rounds $s > t$ that results from changing her bid from her true value $v_{i,t}$ to any other misreport $v'_{i,t}$. Let $Y$ be the event that she bids truthfully in round $t$, and let $\bar{Y}$ be the event that she misreports. Let $u(Y)$ and $u(\bar{Y})$ respectfully be the total utility the bidder achieves in all future rounds conditioned on events $Y$ and $\bar{Y}$. Let $\mathcal{S}$ be the set of all possible outcomes (i.e., allocations and payments) from all future rounds of the auction that this bidder may participate in, and let $w(s)$ be the utility the bidder from outcome $s \in \mathcal{S}$. We

can now bound the player's gain in expected utility from lying by bounding the expected value of $u(\bar{Y})$.

$$
\begin{aligned}
\mathbf{E}\left[u(\bar{Y})\right] &= \int_{\mathcal{S}} w(s) \, \mathrm{P}\left[s|\bar{Y}\right] ds \\
&\leq \int_{\mathcal{S}} w(s)(e^{\epsilon} \, \mathrm{P}\left[s|Y\right] + \tfrac{\epsilon}{T}) ds \\
&\leq \int_{\mathcal{S}} w(s)((1+2\epsilon) \, \mathrm{P}\left[s|Y\right] + \tfrac{\epsilon}{T}) ds \\
&= (1+2\epsilon)\mathbf{E}[u(Y)] + \int_{\mathcal{S}} w(s)\tfrac{\epsilon}{T} ds \\
&\leq \mathbf{E}[u(Y)] + 2\epsilon kh + kh\tfrac{\epsilon}{T}.
\end{aligned}
$$

The first inequality follows from the $(\epsilon, \delta)$-DP guarantee of Theorem 12, the second from the fact that $e^{\epsilon} \leq (1+2\epsilon)$ for $\epsilon < 1$, and the final inequality from the fact that each bidder participates in at most $k$ rounds and her maximum utility is any round is $h$, so both $\mathbf{E}[u(Y)]$ and $\int_{\mathcal{S}} w(s)ds$ are upper bounded by $kh$. Thus the maximum change in utility over all future rounds between any two behaviors in the current round is therefore $2\epsilon kh + kh\tfrac{\epsilon}{T} = kh\epsilon(2 + \tfrac{1}{T})$.

Thus, the overall utility the bidder might gain from misreporting in round $t$ is $kh\epsilon(2 + \tfrac{1}{T})$, which converges to $2kh\epsilon$ as $T \to \infty$.

$\square$

## A.4 Revenue Analysis for Algorithm 1

To prove the main revenue gurantee of Utility-Approximate BIC Online Auction presented in Algorithm 1, i.e Theorem 14 of our, we present a few lemmas to bound the difference in revenue obtained in each round and then sum it over the $T$ rounds to bound the average expected revenue.

In each round, Algorithm 1 runs the optimal mechanism for $\mathbf{H}'_t$, but discretized value profiles are sampled from $\mathbf{D}'$. The following lemma uses tools from the previous subsection to show that the expected revenue of running $\mathcal{M}^*_{\mathbf{H}'_t}$ on samples from $\mathbf{H}'_t$ is not much worse that running $\mathcal{M}^*_{\mathbf{D}'}$ on samples from $\mathbf{D}'$.

**Lemma 32.** *With probability at least* $1 - \alpha$, $\left|Rev(\mathcal{M}^*_{\mathbf{H}'_t}; \mathbf{H}'_t) - Rev(\mathcal{M}^*_{\mathbf{D}'}; \mathbf{D}')\right| \leq 2hn^2\gamma_t$.

*Proof.* We start by re-writing the revenue difference we wish to bound, $\left|\mathrm{Rev}(\mathcal{M}^*_{\mathbf{H}'_t}; \mathbf{H}'_t) - \mathrm{Rev}(\mathcal{M}^*_{\mathbf{D}'}; \mathbf{D}')\right|$ as follows,

$$
\left|(\mathrm{Rev}(\mathcal{M}^*_{\mathbf{H}'_t}; \mathbf{H}'_t) - \mathrm{Rev}(\mathcal{M}^*_{\mathbf{H}'_t}; \mathbf{D}')) + (\mathrm{Rev}(\mathcal{M}^*_{\mathbf{H}'_t}; \mathbf{D}') - \mathrm{Rev}(\mathcal{M}^*_{\mathbf{D}'}; \mathbf{D}'))\right|.
$$

For the first term inside the absolute value, Theorem 24 says that with probability at least $1 - \alpha$, $\left\|D'_j - H'_{j,t}\right\|_{\infty} \leq \gamma_t$ for all $j$, and therefore $D'_j$ and $H'_{j,t}$ are $\gamma_t$-close for all $j$. Applying Theorem 27 gives that $\mathrm{Rev}(\mathcal{M}^*_{\mathbf{H}'_t}; \mathbf{H}'_t) - \mathrm{Rev}(\mathcal{M}^*_{\mathbf{H}'_t}; \mathbf{D}') \leq 3hn^2\gamma_t$ with the same probability. The second term is 0 because $\mathrm{Rev}(\mathcal{M}^*_{\mathbf{H}'_t}; \mathbf{D}') \leq \mathrm{Rev}(\mathcal{M}^*_{\mathbf{D}'}; \mathbf{D}')$, since $\mathcal{M}^*_{\mathbf{D}'}$ is the revenue-optimal mechanism for $\mathbf{D}'$. Therefore, we get $\mathrm{Rev}(\mathcal{M}^*_{\mathbf{H}'_t}; \mathbf{H}'_t) - \mathrm{Rev}(\mathcal{M}^*_{\mathbf{D}'}; \mathbf{D}') \leq 2hn^2\gamma_t$.

A symmetric argument using $\mathcal{M}^*_{\mathbf{D}'}$, gives that $\mathrm{Rev}(\mathcal{M}^*_{\mathbf{D}'}; \mathbf{D}') - \mathrm{Rev}(\mathcal{M}^*_{\mathbf{H}'_t}; \mathbf{H}'_t) \leq 2hn^2\gamma_t$, which completes the proof. $\square$

Now we present a result from Devanur et al. [17] (generalized to multiple item auction) which states that discretization of the value space by rounding down to nearest multiple of $\beta$ only reduces the optimal revenue by an additive factor of $J\beta$ for a $J$-item auction. Intuitively, since bids are always rounded down, this can result in a loss of at most $\beta$ revenue from each of the $J$ items.

**Lemma 33** (Devanur et al. [17]). $Rev(\mathcal{M}^*_{\mathbf{D}'}; \mathbf{D}') \geq Rev(\mathcal{M}^*_{\mathbf{D}}; \mathbf{D}) - \beta J$.

Combining these results, we can now bound the expected revenue of our mechanism for a fixed round.

**Lemma 34.** *With probability at least* $1 - \alpha$, *the expected revenue obtained by Algorithm 1 in the* $t^{th}$ *round,* $Rev(\mathcal{M}^*_{\mathbf{H}'_t}; \mathbf{D}')$, *satisfies,*

$$
Rev(\mathcal{M}^*_{\mathbf{H}'_t}; \mathbf{D}') \geq Rev(\mathcal{M}^*_{\mathbf{D}}; \mathbf{D}) - \beta J - 4hn^2\gamma_t,
$$

*for* $\gamma_t = \sqrt{\frac{\log \frac{n}{\alpha}}{2t}} + \frac{\sigma}{t}\sqrt{\log\frac{h}{\beta}\log T}\sqrt{2\log\left(\frac{2hn}{\beta\alpha}\right)}$ *and* $\sigma = \frac{8\log T \log\frac{h}{\beta}}{\epsilon}\sqrt{\ln\frac{T\log T\log\frac{h}{\beta}}{\epsilon}}$.

*Proof.* Using Lemma 32 gives,

$$\text{Rev}(\mathcal{M}^*_{\mathbf{H}'_t}; \mathbf{D}') \geq Rev(\mathcal{M}^*_{\mathbf{D}'}; \mathbf{D}') - 4hn^2\gamma_t,$$

and applying Lemma 33 gives,

$$\text{Rev}(\mathcal{M}^*_{\mathbf{H}'_t}; \mathbf{D}') \geq Rev(\mathcal{M}^*_{\mathbf{D}}; \mathbf{D}) - \beta J - 4hn^2\gamma_t.$$

$\square$

Now that we have bounded the expected revenue in a fixed round $t$, we can bound the average revenue over $T$ rounds with a union bound over all the rounds to obtain a guarantee for the avaerage expect reveunue of our Algorithm 1 in Theorem 14.

## B   Omitted Proofs and Subroutines for Bid-approximate BIC online auctions (Algorithm 2)

### B.1   Subroutines used in Algorithm 2

#### B.1.1   Strictly Truthful mechanism

To ensure that underbidding is costly for the bidders in the current round $t$, with some probability $\rho_t$, we use a naïve mechanism which is strictly truthful. We simply select a random subset $S$ of $J$ bidders and charge them a uniformly random price $p$ between 0 and $h$. If bidder $i$ is selected in the set $S$ and has bid $b_{i,t}$ is above $p$, they get the item and pay $p$. Otherwise, they pay 0 and are not allocated an item.

---

**Algorithm 4:** StrictlyTruthful

---

**Input:**   Bid profile $\mathbf{b}_t$
Select a subset $S \subseteq [n]$ of size $J$ uniformly at random
Select a price $p \in [0, h]$ uniformly at random
**for** *Each* $s \in S$ **do**
  $\mid$   **if** $b_{s,t} \geq p$ **then** allocate item to $s$ and charge payment $p$ ;
**end**

---

As the maximum possible revenue in a single round is $hJ$, the expected loss in revenue from Algorithm 4 in round $t$ of the mechanism is at most $\rho_t hJ$.

#### B.1.2   Private Payments from Black Box Payments

We now describe how to privately compute payments to charge buyers in round $t$. Our goals for this payment computation are first, to ensure our round $t$ mechanism charges payments according to Theorem 2 (ensuring incentive compatibility for the one-round game), and second, for player $i$'s round $t$ payment to be differentially private in all other round $t$ bidders' bids.

In order to achieve the latter goal of private payments, we select the round $t$ payment for each winning bidder $i$ as a function of $b_{i,t}$, their bid, the private estimated distributions of other bidder types $\mathbf{H}'_t$, and the allocation algorithm $\mathcal{A}$, but *not* as a function of the other bidders' round $t$ bids. This ensures the payment of bidder $i$ is 0 jointly differentially private in round $t$ bids, and $(\epsilon, \delta)$-differentially private in all previous rounds' bids. Concretely, we use black box payments (Algorithm 5) introduced in Archer et al. [2].

Critically, bidder $i$'s payment does not depend on the bid of any other player, save for the extent to which the allocation is affected by other the other players' bids. For a differentially private allocation rule, this implies that player $i$'s payment is differentially private in all other players' bids. $i$ therefore learns almost nothing about other players' behavior from their payment in round $t$.

---

**Algorithm 5:** Black box payments

---

**Input:** Value distribution $\mathbf{H}'_t$, Allocation algorithm $\mathcal{A}$, agent $i$'s bid $b_{i,t}$

**if** *algorithm $\mathcal{A}$ does not allocate to bidder $i$* **then** agent $i$'s payment $p_i \leftarrow 0$;

**else**

    Choose $b'_i$ uniformly from $[0, b_{i,t}]$

    Draw $\mathbf{b}'_{-i} \sim H'_{-i}$ and run $\mathcal{A}(b'_i, \mathbf{b}'_{-i})$

    **if** *algorithm $\mathcal{A}$ allocates to bidder $i$ in the previous step* **then** $X \leftarrow b_{i,t}$;

    **else** $X \leftarrow 0$;

    **if** $X \neq 0$ **then**

        repeatedly draw values $\mathbf{b}'_{-\mathbf{i}} \sim \mathbf{H}'_{-\mathbf{i},\mathbf{t}}$ and run $\mathcal{A}(b'_i, \mathbf{b}'_{-i})$ until the algorithm allocates to

        player $i$, and let $L$ be the number of iterations required

    **end**

    Agent $i$'s payment $p_{i,t} \leftarrow b_{i,t} - L \cdot X$

**end**

---

We make one remark about this variant of black-box payments giving the formal guarantees we will use to analyze our mechanism. These black-box payments are computed from the (strategically) shaded, differentially private estimate of the empirical bid distribution, rather than the true, unaltered distribution, since (a) the empirical, shaded distribution is the best estimate the mechanism has, and (b) maintaining only a differentially private estimate of this distribution ensures that no single bidder's prior rounds' behavior significantly change these payments.

We now present the lemmas about Algorithm 5 which help us analyze the revenue and (approximate) incentive compatibility of the overall mechanism.

Fixing the distribution $\mathbf{H}'_t$, the expected payment for bidder $i$ given her value $b_{i,t}$ will be $p_{i,t}(b_{i,t}) = \mathbf{E}_{\mathbf{b}_{-i}}[p_{i,t}(b_{i,t}, \mathbf{b}_{-i})]$, and the probability she is allocated an item using the allocation mechanism $\mathcal{A}$ is $x_{i,t}(b_{i,t}) = \mathbf{E}_{\mathbf{b}_{-i}}[x_{i,t}(b_{i,t}, \mathbf{b}_{-i})]$. We have the following lemma.

**Lemma 35** (Archer et al. [2]). *Fixing the distribution $\mathbf{H}'_t$ and the allocation mechanism $\mathcal{A}$, the expected payment $p_{i,t}(b_{i,t})$ returned by Black Box payments (Algorithm 5) satisfies*

$$p_{i,t}(b_{i,t}) = b_{i,t} \cdot x_{i,t}(b_{i,t}) - \int_0^{b_{i,t}} x_{i,t}(u) du$$

This property about the payments calculated by Black Box payments directly helps us to translate the expected virtual surplus from the allocation rule i.e the Exponential Mechanism to the expected revenue.

**Lemma 36.** *Fix a distribution $\mathbf{H}'_t$. Then, allocating according to the exponential mechanism $\mathcal{M}_E$ and charging black-box payments yields expected revenue on $\mathbf{H}'_t$ equal to the expected virtual surplus of the allocation selected by $\mathcal{M}_E$.*

*Proof.* The exponential mechanism is monotone, by Lemma 40. Black-box payments are designed to satisfy the payment identity, and so the two together have expected revenue equal to their expected virtual surplus, by Theorem 2. □

### B.1.3 Private, efficient allocation via the Exponential mechanism

We now describe the exponential mechanism, and how we use it to select an allocation in round $t$. The primary difference between this choice and many other uses of the exponential mechanism is that we find a way to select allocations with it in a computationally tractable way (details to follow). Let $\mathcal{X}$ be the set of possible allocations. For any mechanism $\mathcal{M}$, we need an allocation rule $\mathcal{A}$, a function that takes as input the bid profile $\mathbf{b}$, the value distribution $H'$, and returns an allocation $\mathbf{x}(\mathbf{b}) \in \mathcal{X}$. $\mathcal{A}$ can be a randomized algorithm, in which case $x_i(\mathbf{b})$ is a random variable indicating the probability of allocation.

In general, the exponential mechanism runs in time polynomial in the number of elements it selects amongst. If one is using it to select an allocation of $J$ items to some set of bidders, this will often result in exponential runtime.

We use the exponential mechanism to select an allocation in a slightly nonstandard way, in that the quality score we use to measure the utility of outcomes is only an approximation of what we would ideally like to maximize.

We use the private, learned distributions to define estimated virtual values for each bid, and use those estimated virtual values as the arguments to the exponential mechanism. Let $\hat{\phi}(\mathbf{b})$ be the virtual value profile given as input to the exponential mechanism. The exponential mechanism is parameterized by a *quality score* $Q(\hat{\phi}(\mathbf{b}), \mathbf{x})$, a function mapping inputs and outputs of the desired optimization task to a measurement of the outcome's approximation to the optimization task on the input. In our mechanism, we select the quality score for for the Exponential Mechanism $\mathcal{M}_E$ to be the estimated virtual welfare of the allocation on the reported bids:

$$Q(\hat{\phi}(\mathbf{b}), \mathbf{x}) = \sum_{i=0}^{n} \hat{\phi}_i(b_i) x_i.$$

If $\mathbf{b}$ and $\mathbf{b}'$ are two neighboring bid profiles, i.e they differ in at most one bidder's bid, we define the sensitivity $\Delta$ of the quality score as maximum change in quality score over all pairs of neighbouring bid profiles $\mathbf{b}$ and $\mathbf{b}'$ and all possible feasible allocations:

$$\Delta = \max_{\mathbf{x} \in \mathcal{X}, \mathbf{b}, \mathbf{b}'} \left| Q(\hat{\phi}(\mathbf{b}), \mathbf{x}) - Q(\hat{\phi}(\mathbf{b}'), \mathbf{x}) \right|$$

Given a virtual value profile $\phi(\mathbf{b})$ the Exponential Mechanism $\mathcal{M}_E(\mathcal{X}, Q, \phi(\mathbf{b}), \epsilon)$ chooses an allocation $\mathbf{x} \in \mathcal{X}$ with probability

$$\mathrm{P}\left[\mathcal{M}_E(\mathcal{X}, Q, \hat{\phi}(\mathbf{b}), \epsilon) = \mathbf{x}\right] \propto \exp \frac{\epsilon Q(\hat{\phi}(\mathbf{b}), \mathbf{x})}{\Delta} \leq h$$

where the upper bound follows from the fact that as one bidder changing her bid can affect the revenue of any allocation by at most $h$. Since expected revenue is equal to the expected virtual welfare, then the exponential mechanism designed to maximize expected estimated virtual welfare on the reported bids will also approximately maximize expected revenue. We now state the main theorem about the exponential mechanism we will use in what follows.

**Theorem 37** (McSherry and Talwar [33]). *The Exponential Mechanism $\mathcal{M}_E(\mathcal{X}, Q, \hat{\phi}(\mathbf{b}), \epsilon)$ is $\epsilon$-differentially private, and for any $\alpha > 0$,*

$$Pr[Q(\hat{\phi}(\mathbf{b}), \mathcal{M}_E(\mathcal{X}, Q, \hat{\phi}(\mathbf{b}), \epsilon)) \geq \max_{\mathbf{x} \in \mathcal{X}} Q(\hat{\phi}(\mathbf{b}), \mathbf{x}) - \frac{\ln|\mathcal{X}|}{\epsilon} - \frac{h}{\epsilon} \ln\left(\frac{1}{\rho}\right)] \geq 1 - \rho$$

This result holds with high probability for a value bid profile $\mathbf{b}$. Below we state a common corollary of this statement, which allows us to quantify the expected revenue loss over the mechanism's randomness:

**Lemma 38.** *For any virtual bid profile $\hat{\phi}(\mathbf{b})$ and $\rho \in (0, 1)$*

$$\max_{\mathbf{x} \in \mathcal{X}} Q\left(\hat{\phi}(\mathbf{b}), \mathbf{x}\right) - \mathbf{E}_{\mathcal{M}_E}\left[Q(\hat{\phi}(\mathbf{b}), \mathcal{M}_E(\mathcal{X}, Q, \hat{\phi}(\mathbf{b}), \epsilon))\right] \leq \frac{J \ln n}{\epsilon} + \frac{h}{\epsilon} \ln \frac{1}{\rho} + \rho J h.$$

*Proof.* Using Theorem 37, we have

$$max_{\mathbf{x} \in \mathcal{X}} Q\left(\hat{\phi}(\mathbf{b}), \mathbf{x}\right) - \mathbf{E}_{\mathcal{M}_E}\left[Q(\hat{\phi}(\mathbf{b}), \mathcal{M}_E(\mathcal{X}, Q, \hat{\phi}(\mathbf{b}), \epsilon))\right]$$
$$\leq (1 - \rho)\left(\frac{\ln|\mathcal{X}|}{\epsilon} + \frac{h}{\epsilon} \ln \frac{1}{\rho}\right) + \rho J h$$
$$\leq \frac{J \ln n}{\epsilon} + \frac{h}{\epsilon} \ln \frac{1}{\rho} + \rho J h$$

$\square$

The trivial way to implement $\mathcal{M}_E(\mathcal{X}, Q, \hat{\phi}(\mathbf{b}), \epsilon)$ would be to iterate over all $x \in \mathcal{X}$ to calculate $Q(\hat{\phi}(\mathbf{b}), \mathbf{x})$.

Since we are running a $J$-item auction in each round $t$, our feasible allocations are the set

$$\mathcal{X} = \{\mathbf{x} : \mathbf{x} \in \{0, 1\}^n, \|\mathbf{x}\|_1 = J\},$$

which has size $|\mathcal{X}| = \binom{n}{J} \leq n^J$. Thus, explicitly iterating over $\mathcal{X}$ would result in an $\mathcal{O}\left(n^J\right)$ algorithm.

---

**Algorithm 6:** Poly-time Exponential Mechanism for selecting an allocation of $J$ goods $(\mathcal{M}_{polyE}(\phi(\mathbf{b}), \epsilon))$

---

**Parameters:** Privacy parameter $\epsilon$, Sensitivity $\Delta$, Number of bidders to select $J$
**Input:** Virtual bid profile $\hat{\phi}(\mathbf{b})$
$S_0 \leftarrow \{\}$
**for** $i = 1, \cdots, n$ *and* $|S| < J$ **do**

$$\alpha_i = \frac{\exp(\frac{\epsilon \hat{\phi}_i(b_i)}{\Delta}) \sum_{A \in \binom{\{i+1,\cdots,n\}}{J-|S_{i-1}|-1}} (\exp(\frac{\epsilon}{\Delta} \sum_{a \in A} \hat{\phi}_a(b_a)))}{\sum_{B \in \binom{\{i+1,\cdots,n\}}{J-|S_{i-1}|}} (\exp(\frac{\epsilon}{\Delta} \sum_{b \in B} \hat{\phi}_b(b_b)))}$$

With probability $\alpha_i$ **do** $S_i \leftarrow S_{i-1} \cup \{i\}$ **else** $S_i \leftarrow S_{i-1}$

**end**
Return binary allocation $\mathbf{x}$ such $x_i = 1$ for all $i \in S_n$

---

However, since our feasible set $\mathcal{X}$ has a very simple structure (a set of $n$ length binary vector with $\leq J$ ones), we can exploit this structure to select a subset of size $J$ in polynomial time by carefully calculating the marginal probabilities of selecting individual bidders such that the resulting joint probabilities of the subsets is same as in $\mathcal{M}_E(\mathcal{X}, Q, \phi(\mathbf{b}), \epsilon)$.

To take advantage of this structure, we calculate the marginal probability of selecting bidder $i$ conditioned on the bidders we have already sampled or rejected. Calculating these marginal probability can be done using a simple dynamic programming solution in $\mathcal{O}(nJ)$. We select or reject each bidder in turn, leading to a total running time of for Algorithm 6 of $\mathcal{O}(n^2 J)$. We now prove that the allocation selected by Algorithm 6 has the same distribution as $\mathcal{M}_E(\mathcal{X}, Q, \phi(\mathbf{b}), \epsilon)$. Let the mechanism defined in Algorithm 6 be $\mathcal{M}_{polyE}(\phi(\mathbf{b}), \epsilon)$.

**Lemma 39.** $\mathrm{P}\left[\mathcal{M}_{polyE}(\hat{\phi}(\mathbf{b}), \epsilon) = \mathbf{x}\right] = \mathrm{P}\left[\mathcal{M}_E(\mathcal{X}, Q, \hat{\phi}(\mathbf{b}), \epsilon) = \mathbf{x}\right] = \frac{\exp \frac{\epsilon Q(\hat{\phi}(\mathbf{b}), \mathbf{x})}{\Delta}}{\sum_{\mathbf{x}' \in \mathcal{X}} \exp \frac{\epsilon Q(\hat{\phi}(\mathbf{b}), \mathbf{x}')}{\Delta}}$

*Proof.* Let

$$\mathrm{P}[\mathbf{x}] = \frac{\exp \frac{\epsilon \sum_{i=0}^n \hat{\phi}_i(b_i) x_i}{\Delta}}{\sum_{\mathbf{x}' \in \mathcal{X}} \exp \frac{\epsilon \sum_{i=0}^n \hat{\phi}_i(b_i) x_i'}{\Delta}}.$$

Note that $\alpha_i$ in Algorithm 6 is the marginal conditional probability $\mathrm{P}[x_i = 1 | x_1, \cdots, x_{i-1}]$. Thus we set $x_i = 1$ with the corresponding conditional marginal. We conclude the proof using the chain rule of joint probability distributions, which states that

$$\mathrm{P}[x_1, x_2, \cdots, x_n] = \mathrm{P}[x_1] \mathrm{P}[x_1|x_2] \cdots \mathrm{P}[x_n|x_1, \cdot, x_{n-1}].$$

$\square$

The following is a well-understood fact, but we include a short proof for completeness.

**Fact 40.** *For downward-closed environments, the Exponential Mechanism instantiated with any monotone virtual value function as its quality score has a monotone allocation rule.*

*Proof of 40.* Fix the behavior $\mathbf{v}_{-i}$ of players other than $i$. The probability the Exponential Mechanism selects allocation $a \in \mathcal{X}$ is

$$\frac{e^{(\epsilon \sum_{i \in a} \phi_i(v_i))/\Delta}}{\sum_{a' \in \mathcal{X}} e^{(\epsilon \sum_{i \in a'} \phi_i(v_i))/\Delta}}$$

We can view allocation $a \in \mathcal{X}$ as a set of bidders who will receive the item. Then, the probability that player $i$ is in the allocated set is,

$$\sum_{a \ni i} \frac{e^{(\epsilon \sum_{i' \in a} \phi_{i'}(v_{i'}))/\Delta}}{\sum_{a' \in \mathcal{X}} e^{(\epsilon \sum_{i' \in a'} \phi_{i'}(v_{i'}))/\Delta}}$$

thus, the numerator increases for each allocation $a$ containing $i$ because $\phi_i$ is monotone, and because $\frac{a+x}{a+x+y} > \frac{x}{x+y}$, this probability increases in $v_i$. $\square$

Now recall that we defined $Rev(\mathcal{M}; D)$ to be the expected revenue generated by the mechanism $\mathcal{M}$ on a value distribution $D$. Let the mechanism we use in round $t$ for allocation and pricing be $\mathcal{M}_{\mathbf{H'}_t}$ and let $\mathcal{M}^*_{\mathbf{H'}_t}$ be the optimal mechanism on $\mathbf{H'}_t$. Using Lemma 36, we can see that $Rev(\mathcal{M}^*_{\mathbf{H'}_t}; \mathbf{H'}_t) = \mathbf{E}_{\mathbf{b} \sim \mathbf{H'}_t}[max_{\mathbf{x} \in \mathcal{X}} Q(\phi_{\mathbf{t}}(\mathbf{b}), \mathbf{x})]$ where $\phi_{\mathbf{t}}(\mathbf{b})$ calculates the virtual bid using $\mathbf{H'}_t$ as prior. Combining this guarantees with the guarantee over the virtual surplus given by exponential mechanism, we get the following lemma.

**Lemma 41.** $Rev(\mathcal{M}_{\mathbf{H'}_t}; \mathbf{H'}_t) \geq Rev(\mathcal{M}^*_{\mathbf{H'}_t}; \mathbf{H'}_t) - (\frac{J \ln n}{\epsilon} + \frac{h}{\epsilon} \ln \frac{1}{\rho_t} + \rho_t J h) - \rho_t h J.$

*Proof.* The lemma is directly a consequence of taking expectation of over $\mathbf{b}$ in Lemma 38 with respect to $\mathbf{H'}_t$ with $\rho = \rho_t$ and then using Lemma 36. The last term $\rho_t h J$ comes form the fact in round $t$, we use the strictly truthful mechanism (Algorithm 4) with probability $\rho_t$ and lose up to $hJ$. $\qquad\square$

## B.2 Differentially Private Bid Distribution Estimation

We now present a lemma which compares how close our differentially private empirical bid distribution estimates are to the true bid distributions, under this assumption that the bid distributions do not change too quickly.

**Lemma 42.** *Assume we have a sequence of distributions $F'_{i,t}$ with support on $[0, h]$, and assume that for every $t$ we have $\|F'_{i,t} - F'_{i,t+1}\|_\infty \leq \lambda_t$ for some sequence $\lambda_1, \lambda_2, \ldots$ satisfying $\Sigma_T := \sum_{t=1}^{T-1} \frac{t\lambda_t}{T} = o(1)$. Let $b_{i,t} \sim F'_{i,t}$ and let $H_{i,T}$ be the empirical distribution of $b_{i,1}, \ldots, b_{i,T}$. Then with probability at least $1 - \alpha$ we have*

$$\|H_{i,T} - F'_{i,T}\|_\infty \leq \sqrt{\frac{\log(2h/(\beta\alpha))}{2T}} + \Sigma_T.$$

*Proof.* Let $\bar{F}$ be the distribution constructed from the uniform mixture of the distributions $\mathbf{F}_{i.1}, \ldots, F'_{i,T}$, that is $\bar{F} := \frac{1}{T} \sum_{t=1}^T F'_{i,t}$. Observe that, via repeated application of the triangle inequality,

$$
\begin{aligned}
\|H_{i,T} - F'_{i,T}\|_\infty &\leq \|H_{i,T} - \bar{F}\|_\infty + \|\bar{F} - F'_{i,T}\|_\infty \\
&\leq \|H_{i,T} - \bar{F}\|_\infty + \frac{1}{T} \sum_{t=1}^T \|F'_{i,t} - F'_{i,T}\|_\infty \\
&\leq \|H_{i,T} - \bar{F}\|_\infty + \frac{1}{T} \sum_{t=1}^{T-1} \sum_{j=t}^{T-1} \|F'_{i,j} - \mathbf{F}_{i.j+1}\|_\infty \\
&\leq \|H_{i,T} - \bar{F}\|_\infty + \frac{1}{T} \sum_{t=1}^{T-1} \sum_{j=t}^{T-1} \lambda_j = \|H_{i,T} - \bar{F}\|_\infty + \Sigma_T.
\end{aligned}
$$

Thus, to complete the proof, we must simply show that with probability at least $1 - \alpha$ that $\|H_{i,T} - \bar{F}\|_\infty \leq \sqrt{\frac{\log(2h/(\beta\alpha))}{2T}}$. To do this, first select some $u \in [0, h]$ and notice that

$$\mathbf{E}[H_{i,T}(u)] = \mathbf{E}\left[\frac{1}{T} \sum_{t=1}^T \mathbf{1}[b_{i,t} \leq u]\right] = \frac{1}{T} \sum_{t=1}^T \mathrm{P}[b_{i,t} \leq u] = \frac{1}{T} \sum_{t=1}^T F'_{i,t}(u) = \bar{F}(u);$$

in other words, $H_{i,T}(u)$ is an unbiased estimator of $\bar{F}(u)$. Since each indicator variable $\mathbf{1}[b_{i,t} \leq u]$ is bounded in $[0, 1]$, we may apply Hoeffding's Inequality to obtain

$$\mathrm{P}\left[\left|\frac{1}{T} \sum_{t=1}^T \mathbf{1}[b_{i,t} \leq u] - \bar{F}(u)\right| > \xi\right] \leq 2 \exp(-2T\xi^2). \tag{3}$$

Observe that, since $H_{i,T}$ and $\bar{F}$ are step functions which only change at points $U = \{\beta, 2\beta, \ldots, \frac{h}{\beta}\beta\}$, it holds for any $\xi > 0$ that

$$\|H_{i,T} - \bar{F}\|_\infty \geq \xi \Longleftrightarrow \exists u \in [0, h] : |H_{i,T}(u) - \bar{F}(u)| \geq \xi \Longleftrightarrow \exists u \in U : |H_{i,T}(u) - \bar{F}(u)| \geq \xi.$$

Thus we have that

$$
\begin{aligned}
\mathrm{P}\left[\left\|H_{i,T} - \bar{F}\right\|_\infty \geq \xi\right] &= \mathrm{P}\left[\exists u \in U : \left|\frac{1}{T}\sum_{t=1}^{T}\mathbf{1}[b_{i,t} \leq u] - \bar{F}(u)\right| > \xi\right] \\
&\leq \sum_{u \in U}\mathrm{P}\left[\left|\frac{1}{T}\sum_{t=1}^{T}\mathbf{1}[b_{i,t} \leq u] - \bar{F}(u)\right| > \xi\right] \\
\text{(using (3))} \quad &\leq \frac{2h}{\beta}\exp(-2T\xi^2).
\end{aligned}
$$

The proof is completed by setting $\alpha$ equal to the final expression and solving for the value $\xi$. $\qquad\square$

The next theorem states that our differentially private estimates of $H_{i,t}$ are fairly close to the true distributions.

**Theorem 43.** *After $t$ rounds Algorithm 2, it holds with probability at least $1 - \alpha$ that*

$$
\left\|H'_{i,t} - F'_{i,t}\right\|_\infty \leq \gamma_t \quad \text{for every } i \in [n] ,
$$

*where $\gamma_t = \sqrt{\frac{\log(2hn/\beta\alpha)}{2t}} + \Sigma_t + \frac{\sigma}{t}\sqrt{\log\frac{h}{\beta}\log T}\sqrt{\log\left(\frac{hn}{\beta\alpha}\right)}$ and $\sigma = \frac{8\log T\log\frac{h}{\beta}}{\epsilon}\sqrt{\ln\frac{\log T\log\frac{h}{\beta}}{\delta}}$.*

*Proof.* We assume the same $\Sigma_t$ for each bidder type. Next, apply a union bound on Lemma 42 for all players and use the triangle inequality with Lemma 23. The result is analogous to Theorem 24. $\quad\square$

The next lemma states that the distribution over critical prices induced by our learned distribution is close to that induced by the true bid distribution.

**Lemma 44.** *For a bidder $i$, let $G_{i,t}$ be the distribution of critical prices offered to $i$ when the distribution of other bidders is $\mathbf{F}'_{-i,t}$ and let $G'_{i,t}$ be the distribution of critical prices when the distribution of other bidders is $\mathbf{H}'_{-i,t}$, then $\left\|G_{i,t} - G'_{i,t}\right\|_\infty \leq (n-1)\gamma_t$.*

*Proof.* From Theorem 43, we have that $\mathbf{F}'_{-i,t}$ and $\mathbf{H}'_{-i,t}$ are $\gamma_t$ close. The exponential mechanism is a monotone allocation rule (Fact 40). The lemma then follows from Lemma 29. $\quad\square$

### B.3 Truthfulness Guarantees of Algorithm 2

**Lemma 45.** *Let $r_{i,t}(v)$ be the true BIC payment (from Theorem 2) for bidder $i$ in round $t$ when value is $v$ and let $p_{i,t}(v)$ be the expected payment charged by Algorithm 1 by using Black Box payments (Algorithm 5), then $|r_{i,t}(v) - p_{i,t}(v)| \leq 2hn\gamma_t$.*

*Proof.* The proof follows from the fact the Black Box payments in round $t$ calculate the BIC in expectation with respect to $H'_{i,t}$ and from using Lemma 30. $\quad\square$

We now use these results to prove that our Online DP auction satisfies differential privacy.

**Lemma 46.** *Fix a round $t$ and bidder population $i$. Then, consider the case when the mechanism runs the exponential allocation and black-box payments. Then, truthful reporting for the bidder from population $i$ present in round $t$ earns within $6hn\gamma_t$ of the optimal utility they could achieve.*

Recall that $r_{i,t}(v)$ is the true BIC payment and $p_{i,t}(v)$ be the expected payment charged by Algorithm 1. Let $v_{i,t}$ be the true value of the bidder $i$ in round $t$, and let $v'$ be some misreport. Since $r_{i,t}(v)$ is the correct $BIC$ payments, the maximum utility bidder $i$ could have received if the payments charged were BIC be $u(v_{i,t}) = v_{i,t}x_{i,t}(v_{i,t}) - r_{i,t}(v_{i,t}) \geq u(v')x_{i,t}(v') - r_{i,t}(v')$, but the actual utility achieved by the bidder in the current as a function of their bid is $u'(v) = vx_{i,t}(v) - p_{i,t}(v)$.

From Lemma 45, we have that $|r_{i,t}(v) - p_{i,t}(v)| \le 2hn\gamma_t$. So if the bidder bids $v'$ instead of $v_{i,t}$, then the utility achieved by $i$ in round $t$ is

$$
\begin{aligned}
u'(v') &= v' x_{i,t}(v') - p_{i,t}(v') \\
&\le v' x_{i,t}(v') - r_{i,t}(v') + 2hn\gamma_t \\
&\le v_{i,t} x_{i,t}(v_{i,t}) - r_{i,t}(v_{i,t}) + 2hn\gamma_t \\
&\le v_{i,t} x_{i,t}(v_{i,t}) - p_{i,t}(v_{i,t}) + 4hn\gamma_t \\
&= u'(v_{i,t}) + 4hn\gamma_t.
\end{aligned}
$$

The next lemma shows that a bidder cannot gain a lot in future rounds by manipulating her bid in the current round.

**Lemma 47.** *In any round $t$, for bidder $i$ with value $v_{i,t}$, let $u_i(w)$ be their expected utility in the current round as a function of their bid $w$. If they report bid $b_{i,t}$ such that $b_{i,t} \le v_{i,t} - \eta$, then*

$$
u_i(b_{i,t}) \le u_i(v_{i,t}) - \frac{\rho_t J \eta^2}{2hn} + 4hn\gamma_t.
$$

*Proof.* With probability $\rho_t$ we use algorithm 4, and if we use algorithm 4 then with proability $\frac{J}{n}$, the bidder $i$ is selected for allocation is offered a random price $p \in [0, h]$. If $b_{i,t} = v_{i,t} - y$ where $y \ge \eta$, then with probability at least $\frac{\eta}{h}$, $p \in [b_{i,t}, v_{i,t}]$, i.e bidder $i$ loses utility in this case, which they could have gained if they report their value truthfully. If $p \in [b_{i,t}, v_{i,t}]$, and bidder $i$ is selected then the utility lost by bidder $i$ by not bidding truthfully is $v_{i,t} - p$. With probability $(1 - \rho_t)$, we use exponential mechanism and black box payments from lemma 46, the gain in utility from underbidding in this case be at most $4hn\gamma_t$. Combining the two cases, we can get a lower bound on the loss of utility in the current round

$$
\begin{aligned}
u_i(v_{i,t}) - u_i(b_{i,t}) &\ge \frac{\rho_t J}{n} \left( \int_{b_{i,t}}^{v_{i,t}} (v_{i,t} - p) \frac{1}{h} dp \right) - (1 - \rho_t) 4hn\gamma_t \\
&\ge \frac{\rho_t J (v_{i,t} - b_{i,t})^2}{2hn} - 4hn\gamma_t \\
&\ge \frac{\rho_t J \eta^2}{2hn} - 4hn\gamma_t.
\end{aligned}
$$

$\square$

Lemma 47 shows that that the bidder can lose utility in the current round if they underbid because of the penalty imposed by the strictly truthful mechanism (Algorithm 4).

Now we have bounded all three sources of change in utility for shading one's bid: how much they can gain from future rounds if they underbid today, how much they are penalized by Algorithm 4 for underbidding, and how much they can gain from current round by underbidding. Combing these three results, we can now prove that the mechanism in round $t$ is $\eta_t$-bid approximate BIC (Definition 7).

*Proof for Theorem 19.* If the $v_{i,t} - b_{i,t} > \eta_t = h\sqrt{\frac{8n^2\gamma_t + 6k\epsilon}{\rho_t J}}$, then from Lemma 47 the loss in utility in current round is

$$
\ge \frac{\rho_t J \eta_t^2}{2hn} - 4hn\gamma_t = 4hn\gamma_t + 3kh\epsilon - 4hn\gamma_t \ge kh\epsilon \left(2 + \frac{1}{T}\right)
$$

and from Lemma 17, $kh\epsilon(2 + \frac{1}{T})$ is the maximum gain in utility the player can achieve in future rounds by underbidding. Hence any bid $b_{i,t} < v_{i,t} - \eta_t$ cannot be an equilibrium bid because the total gain in utility in future rounds from underbidding in the current round is less than the loss in utility in current round because of underbidding by more than $\eta_t$. $\square$

## B.4 Revenue Analysis for Algorithm 2

In this section, we bound the expected revenue achieved by our algorithm in some exact equilibrium compared to the optimal expected revenue facing truthful bids, $\text{Rev}(\mathcal{M}_{\mathbf{D}}^*; \mathbf{D})$. In Algorithm 2, with probability $\rho_t$, we execute the strictly truthful mechanism (Algortithm 4) which is also has monotone allocation rule. As stated earlier, let the mechanism we run in round $t$ be $\mathcal{M}_{\mathbf{H}'_t}$, it is clear that the allocation rule used in $\mathcal{M}_{\mathbf{H}'_t}$ is monotone. In round $t$, the bids received by Algorithm 2 come from $\mathbf{F}'_t$. Let $\mathcal{M}_{\mathbf{F}'_t}^*$ be the optimal mechanism on $\mathbf{F}'_t$. As we have shown that our Differentially private estimate is close to $\mathbf{F}'_t$, we can use the theory developed in subsection $A.2$ to bound the revenues. Particularly, we have the following lemma.

**Lemma 48.** *With probability at least* $1 - \alpha$, $|Rev(\mathcal{M}_{\mathbf{H}'_t}; \mathbf{H}'_t) - Rev(\mathcal{M}_{\mathbf{H}'_t}; \mathbf{F}'_t)| \leq 2hn^2\gamma_t$.

*Proof.* Theorem 43 implies that for all $j$, $\left\|F'_{j,t} - H'_{j,t}\right\|_\infty \leq \gamma_t$; As $\mathcal{M}_{\mathbf{H}'_t}$ has a monotone allocation rule, applying Theorem 27 completes the proof. $\square$

We showed that $\mathcal{M}_{\mathbf{H}'_t}$ has similar revenue on both $\mathbf{F}'_t$ and $\mathbf{H}'_t$, to prove our required bound, we also need to show that the optimal revenue on these distributions is also.

**Lemma 49.** *With probability at least* $1 - \alpha$, $\left|Rev(\mathcal{M}_{\mathbf{H}'_t}^*; \mathbf{H}'_t)) - Rev(\mathcal{M}_{\mathbf{F}'_t}^*; \mathbf{F}'_t)\right| \leq 2hn^2\gamma_t$.

*Proof.* Using similar arguments as 48, $\text{Rev}(\mathcal{M}_{\mathbf{H}'_t}^*; \mathbf{H}'_t)) - \text{Rev}(\mathcal{M}_{\mathbf{H}'_t}^*; \mathbf{F}'_t)) \leq 2hn^2\gamma_t$, and since $\text{Rev}(\mathcal{M}_{\mathbf{H}'_t}^*; \mathbf{F}'_t)) \leq \text{Rev}(\mathcal{M}_{\mathbf{F}'_t}^*; \mathbf{F}'_t)$, we get $\text{Rev}(\mathcal{M}_{\mathbf{H}'_t}^*; \mathbf{H}'_t)) - \text{Rev}(\mathcal{M}_{\mathbf{F}'_t}^*; \mathbf{F}'_t) \leq 2hn^2\gamma^t$. Using same argument on $\mathcal{M}_{\mathbf{F}'_t}^*$, we get $\text{Rev}(\mathcal{M}_{\mathbf{D}'}^*; \mathbf{F}'_t) - \text{Rev}(\mathcal{M}_{\mathbf{H}'_t}^*; \mathbf{H}'_t)) \leq 2hn^2\gamma^t$. $\square$

We recall that $\mathbf{F}'_t$ is the rounded down distribution of the actual bid distribution $\mathbf{F}_t$ in round $t$. Using the same result from [17], we have

**Lemma 50** ([17]). $Rev(\mathcal{M}_{\mathbf{F}'_t}^*; \mathbf{F}'_t) \geq Rev(\mathcal{M}_{\mathbf{F}_t}^*; \mathbf{F}_t) - \beta J$.

In Theorem 19, we show that in round $t$, if the the value for a bidder $i$ is $v_{i,t} \sim D_i$, then the bid $b_{i,t}$ is at most $\eta_t$ less than it which means that the distribution of the bids and the values has a nice coupling which mentioned in the introduction of this section. Recall that $\mathbf{F}_{i,t}$ is the distribution of bids of player $i$ in round $t$. With very similar arguments to Lemma 50 we can show the following.

**Lemma 51.** $Rev(\mathcal{M}_{\mathbf{F}_t}^*; \mathbf{F}_t) \geq Rev(\mathcal{M}_{\mathbf{D}}^*; \mathbf{D}) - \eta_t J$.

*Proof.* The proof follows identically to the proof of Lemma 50 in [17]. $\square$

Now we have all the pieces to bound the expected revenue achieved by Algorithm 2 in a fixed round $t$.

**Theorem 52.** *Using Algorithm 2, in round $t$, with probability at least $1 - \alpha$ the expected revenue $Rev(\mathcal{M}_{\mathbf{H}'_t}; \mathbf{F}'_t)$ achieved by the mechanism satisfies:*

$$\text{Rev}(\mathcal{M}_{\mathbf{H}'_t}; \mathbf{F}'_t) \geq \text{Rev}(\mathcal{M}_{\mathbf{D}}^*; \mathbf{D}) - 4hn^2\gamma_t - \frac{J\ln n}{\epsilon} - \frac{h}{\epsilon}\ln\frac{1}{\rho_t} - (\eta_t + \beta + 2\rho_t h)J$$

*where $\gamma_t = \sqrt{\frac{\log(2hn/\beta\alpha)}{2t}} + \Sigma_t + \frac{\sigma}{t}\sqrt{\log\frac{h}{\beta}\log T}\sqrt{\log\left(\frac{hn}{\beta\alpha}\right)}$ and $\eta_t = h\sqrt{\frac{8n^2\gamma_t + 6k\epsilon}{\rho_t J}}$.*

*Proof.* Using Lemma 41 and 48 and setting we have

$$\text{Rev}(\mathcal{M}_{\mathbf{H}'_t}; \mathbf{F}'_t) \geq \text{Rev}(\mathcal{M}_{\mathbf{H}'_t}^*; \mathbf{H}'_t) - 4hn^2\gamma_t - \frac{J\ln n}{\epsilon} - \frac{h}{\epsilon}\ln\frac{1}{\rho_t} - 2\rho_t hJ,$$

We apply Lemma 49 to get

$$\text{Rev}(\mathcal{M}_{\mathbf{H}'_t}; \mathbf{F}'_t) \geq \text{Rev}(\mathcal{M}_{\mathbf{H}'_t}^*; \mathbf{H}'_t) - 4hn^2\gamma_t - \frac{J\ln n}{\epsilon} - \frac{h}{\epsilon}\ln\frac{1}{\rho_t} - 2\rho_t hJ.$$

Now, applying Lemmas 51 and 50 completes the proof. $\square$

To get a bound on the average expected revenue achieved by Algorithm 2, we sum over all rounds and take union bound.

*Proof of Thorem 20.* Similar to Theorem 14, when we sum terms from Theorem 52, the leading term in $\frac{1}{T}\sum_{t=1}^{T}\gamma_t$, turns out to be $\tilde{O}(\frac{1}{\sqrt{T}})$. Keeping $\rho_t$ as constant $\rho$, $\frac{1}{T}\sum_{t=1}^{T}\eta_t$, can be bounded by $hnJ^{-1/3}(\tilde{O}(\frac{1}{T^{1/4}})) + h\sqrt{\frac{6k\epsilon}{\rho J}}$. Adding the rest of the terms and keeping only leading terms in $T$, we get

$$\frac{1}{T}\sum_{t=1}^{T}\mathrm{Rev}(\mathcal{M}_{\mathbf{H'}_t};\mathbf{F}'_t) \geq \mathrm{Rev}(\mathcal{M}_{\mathbf{D}}^*;\mathbf{D}) - hnJ^{2/3}\tilde{O}(\tfrac{1}{T^{1/4}}) - \tfrac{J\ln n}{\epsilon} - \tfrac{h}{\epsilon}\ln\tfrac{1}{\rho} - hJ(\sqrt{\tfrac{6k\epsilon}{\rho J}}+2\rho) - \beta J$$

Optimizing over $\rho$ and setting it to $\left(\frac{3k\epsilon}{2J}\right)^{1/3}$ gives the theorem. $\qquad\square$

## B.5 Omitted Proofs

*Proof of Lemma 16.* The mechanism makes 3 choices in each round: how to allocate items, how to charge bidders, and how to update the estimated distributions. The distribution estimates are maintained in a $(\epsilon, \epsilon/T)$-private manner, by Theorem 22.

Allocations and payments are chosen in one of two ways: either according to the exponential mechanism, followed by black-box mechanism, or by using the strictly truthful mechanism Algorithm 4. In the former case, the allocation rule is $\epsilon$ differentially private by Theorem 37, and the payment for bidder $i$ is a function of her bid in round $t$ and postprocessing of the privately maintained distribution estimates, and are therefore jointly $\epsilon$ jointly differentially private for bidder $i$ (by Lemma 9). In the latter case, the initial allocation is chosen uniformly at random and therefore perfectly private; buyer $i$'s ultimate allocation and payment is a postprocessing of this perfectly private selection along with her bid, and is therefore perfectly jointly differentially private.

Thus, since privacy guarantees sum under composition, the entire mechanism is $(3\epsilon, 3\epsilon/T)$-jointly differentially private. $\qquad\square$

*Proof of Lemma 17.* This follows from the privacy guarantee of Lemma 16 and an argument identical to Theorem 13 upper-bounding the benefit of misreporting. Thus, the benefit of misreporting in round $t$ to all future rounds in which this bidder participates is at most $\epsilon kh(2 + \frac{1}{T})$. $\qquad\square$