[Reviews · NeurIPS 2019]

Reviewer 1



This paper concerns a repeated multi-unit auction, where a collection of items are sold to bidders each round. The seller uses past bidding behavior to learn how to set a reserve price for the auction. The buyers are aware that the seller is learning, and might participate multiple times, so they might change their behavior to affect future prices. The authors propose a differential privacy (DP) approach to this problem. They show that if the seller maintains their data about the bidding history in a DP way, then truthful bidding each round will be an approximate equilibrium for the agents. If the seller also determines the allocation and payments in a DP manner, then it is possible to get the stronger guarantee that exact equilibrium behavior is "close to" truthful bidding, leading to approximately optimal revenue. There is a lot to like about this paper. The problem of repeated sale to forward-looking bidders is an important one, and a hot topic in the intersection of microeconomics and learning theory. Differential privacy is a natural tool to bring to bear on this problem. The authors do a good job of explaining the connections between DP, known DP algorithms, and their various incentive properties. My main concern is the incremental contribution over Liu et al. (cited as [27]). Paper [27] solves a nearly identical problem. The main difference is that [27] learns an *anonymous* reserve, which is optimal only when the agent distributions are identical; this paper extends their approach to Myerson auctions with per-agent reserves. While this is an important extension, the difference between the papers is not discussed at all aside from a cursory mention. Indeed, from a technical perspective, all the ideas from this paper seem to be present in [27] also, so the impression is that the marginal technical contribution is very minor. The paper would be *much* improved with a thorough discussion of what new ideas/techniques were needed for this extension. A more minor issue is the modeling choice that exactly one bidder from each of the n populations is present in each round. Is this necessary? It feels very artificial. As far as I can tell, the results would go through if agents are instead selected each round from an aggregate population of size N, divided into n sub-populations, and the sub-population of each agent is publicly observable. I.e., sometimes there might be 2 agents from one population and 0 from another. I find this model more natural; do the results carry over to such a setting? Overall, I find the paper hard to judge without a detailed comparison to [27]. I think the extension to optimal auctions is important, but as far as I can tell the main body does not contain much content about the new ideas/techniques needed for this, and mainly goes over ideas already present in [27]. In its current form, I think the paper is a marginal case. [Post Author Response] The proposed paragraph describing how the paper relates to [27] would be very helpful. Though I think it's important to clarify the distinction from [27] not just in terms of problem setup, but also techniques after having reduced to the underlying DP problem. In general, I am convinced post-rebuttal (and having seen the other reviews) that the technical contribution relative to [27] is above the bar. I have raised my score. I do still encourage the authors to emphasize the technical proof differences from [27] (e.g., in the way joint DP is established), as this is currently hard to see from the main body of the paper.

Reviewer 2



The paper considers the problem of learning optimal auctions in repeated interaction with bidders where bidders are not myopic (and they may misreport in an early auction to get a better outcome in a later auction). The authors use tools from differential privacy to learn nearly-optimal auctions of the historical bids. A necessary assumption is that bidders don’t show up too often’’; prior results have ruled out positive results if that’s the case. The authors present 2 algorithms in this case, based on two notions of approximately truthful: the first guarantees that bidding the true value yields approximately optimal utility, while the other guarantees an equilibrium where the bids of all agents are close to their true values. Overall I really like the paper. The setting is well-motivated: it builds on an increasingly large literature on learning optimal auctions, but with the reasonable restriction that the method should still be incentive compatible for non-myopic bidders. The algorithms that are proposed make use of tools from differential privacy, but the analysis is thorough and non-trivial. Small remarks: The authors may be interested in the work by Lehaie et al. “Testing Incentive Compatibility in Display Ad Auctions” and follow-ups which deals with the complementary problem for an advertiser to determine whether their bid is used to set a reserve price in later auctions. Typos: Ln 164: “the an” Ln 186: “non-sanitzed” [edit after author feedback: no new points were raised and thus the review remained the same]

Reviewer 3



The paper is overall well-executed. To my knowledge this approach, while intuitive, has not been explored in detail before. My primary comment is that several important strands of related work are omitted. First, there is a line of work that illustrates why the utility-approximate guarantee is a weak one. In particular, after the McSherry-Talwar paper, people realized that while differentially privacy limits the loss from truthful reporting, it also limits the gains. So there may well be obvious, yet individually small improvements that could undermine the mechanism. To counter this, a stronger guarantee like bid-approximate is needed. Papers in this line include [35] (which is currently only mentioned in passing, “Is Privacy Compatible with Truthfulness” by David Xiao ITCS 2013, and “Truthful Mechanisms for Agents that Value Privacy by Chen et al. ACM TEAC 2016. Second, there are some other recent approaches to this problem other than those cited. See “Selling to a No-Regret Buyer” by Braverman et al. EC 2018 and references therein. Third, there are papers looking at applying differential privacy to advertising systems, one of the examples where this paper is relevant. These include “PriPeARL: A Framework for Privacy-Preserving Analytics and Reporting at LinkedIN” by Kenthapadi and Tran and “A Practical Application of Differential Privacy to Personalized Online Advertising.”

[Author Response · NeurIPS 2019]

We thank the reviewers for their careful examination of our work. All three reviewers had helpful suggestions that we wish to address.

We absolutely agree with R1 that a robust discussion about the differences between our results and Liu et al. '18 is missing from the current draft. While a detailed comparison to Liu et al. was present in an earlier draft, it was removed in shortening the paper for the NeurIPS submission, and we acknowledge that this was an error. Here is a summary of this comparison that will be included in the final draft: Liu et al. show how to learn mechanisms with incentive guarantees, and their results do rely on differential privacy, but our results differ in two significant ways:

1. Our setting takes a Bayesian model of buyer types, where bidder values are drawn from a fixed distribution. Liu et al considered a model where buyers are adversarially selected.

2. Liu et al compare their mechanisms to the optimal single price (or optimal anonymous reserve price, with multiple bidders). In other words, their benchmark comes from a family governed by only a single parameter. The benchmark used in our results is much more challenging: we compare to the Bayes-optimal auction for the unknown distribution from which values are sampled. Generally, second-price auctions with anonymous reserve prices (as used in Liu et al) are suboptimal in our setting. The Bayesian-optimal mechanism we compare to is instead parametrized by a virtual value function for each bidder, a significant increase in complexity.

The Bayesian setting and Bayesian-optimal benchmark force us to adopt very different analysis techniques. Instead of the expert learning approach adopted by Liu et al., our approach is closer to the literature on sample complexity in single-shot mechanism design, and our main technical contribution is showing that introducing noise to the learning process for privacy can be done without significantly harming the revenue of the mechanisms we learn.

We thank R5 for pointing out other relevant work. We agree about the potential weakness of utility-approximate BIC (and that differential privacy doesn't necessarily give something stronger), which is why we included results for bid-approximate BIC in Section 4. Xiao '11 and Nissim et al. '12 rightly pointed this out, and we will gladly cite both in all future versions of our work. Similarly, Braverman et al. give results for nonmyopic buyers, and we will also include this reference in our revisions. Finally, the work relating privacy to online advertising, while sharing keywords with our work, generally is motivated by the idea of personalizing advertising to the viewer, rather than maintaining private estimates of the advertisers' value distributions. We nonetheless appreciate the references.

Finally, we remark that our techniques do indeed apply to the alternative model suggested by R1 (as well as to several other natural modifications). We will gladly incorporate a discussion of other settings in which our results apply.

[Meta-Review · NeurIPS 2019]

The question of understanding and designing auctions for non-myopic buyers is a key one in modern mechanism design and this work makes new and interesting contributions. Although the high level ideas are similar to past work, namely the paper by Liu et al., there is enough novelty in this work as the authors extend it beyond auctions with anonymous reserves.